ARTICLES
## OPEN

# Structural basis of RNA processing by human mitochondrial RNase P

Arjun Bhatta [1,2], Christian Dienemann[3], Patrick Cramer [3,4] and Hauke S. Hillen [1,2,4] ✉

**Human mitochondrial transcripts contain messenger and ribosomal RNAs flanked by transfer RNAs (tRNAs), which are excised by mitochondrial RNase (mtRNase) P and Z to liberate all RNA species. In contrast to nuclear or bacterial RNase P, mtRNase P is not a ribozyme but comprises three protein subunits that carry out RNA cleavage and methylation by unknown mechanisms. Here, we present the cryo-EM structure of human mtRNase P bound to precursor tRNA, which reveals a unique mechanism of substrate recognition and processing. Subunits TRMT10C and SDR5C1 form a subcomplex that binds conserved mitochondrial tRNA elements, including the anticodon loop, and positions the tRNA for methylation. The endonuclease PRORP is recruited and activated through interactions with its PPR and nuclease domains to ensure precise pre-tRNA cleavage. The structure provides the molecular basis for the first step of RNA processing in human mitochondria.**

The mitochondrial DNA (mtDNA) encodes for 13 essential subunits of the respiratory chain and is expressed by dedicated molecular machineries[1]. In humans, mtDNA is transcribed by a mitochondrial RNA polymerase (mtRNAP), which produces polycistronic primary transcripts that contain ribosomal RNAs and messenger RNAs interspersed by transfer RNAs (tRNAs)[2–4]. These transcripts then undergo a number of processing and maturation steps to produce functional RNAs. The first step consists of endonucleolytic excision of the tRNA sequences, which separates the three different RNA species and thus constitutes a key event in mitochondrial gene expression[2–4]. This is carried out by mtRNase P and Z, which cleave at the 5′ and 3′ borders of mitochondrial pre-tRNAs, respectively[5–9].

RNase P enzymes are generally responsible for processing the 5′ ends of pre-tRNAs in all domains of life[10,11]. In most bacteria, archaea and many eukaryotes, they are ribozymes composed of a catalytic RNA and a varying number of accessory protein factors. Some eukaryotic organisms additionally or alternatively contain single-subunit protein-only RNase P enzymes (PRORPs) that act in nuclear and organellar RNA processing[12–14]. Mammalian mitochondrial RNase P is evolutionarily unique because it is a multisubunit protein complex consisting of PRORP (also known as MRPP3), which is homologous to single-subunit PRORPs, and the two additional subunits TRMT10C (also known as MRPP1) and SDR5C1 (also known as MRPP2)[7,15]. This multimeric complex carries out both pre-tRNA cleavage and methylation, and is thus a central player in mitochondrial gene expression.

TRMT10C is a multifunctional protein that plays a role in both RNA processing and tRNA methylation. It belongs to the TRMT10 family of SPOUT-fold methyltransferases, which act as tRNA modifying enzymes in both archaea and eukaryotes[16,17]. They catalyze the *S*-adenosylmethionine (SAM)-dependent N1-methylation of specific residues, either with single-base specificity (m$^1$G or m$^1$A) or dual specificity (m$^1$G and m$^1$A)[18]. TRMT10C consists of an N-terminal domain (NTD) and a dual-specificity C-terminal methyltransferase domain, which are both required for establishing

a conserved m$^1$G/A methylation at position 9 of mitochondrial tRNAs[19,20]. The structure of the TRMT10C methyltransferase domain has been reported, but it is not known how it binds and recognizes its substrate[19]. In contrast to other TRMT10 enzymes, which act as monomers[18], TRMT10C requires the second subunit of mtRNase P, SDR5C1, for efficient pre-tRNA methylation[15]. SDR5C1 is a dehydrogenase involved in both metabolic processes and organellar RNA processing[7,21]. It forms a homotetramer in which each monomer adopts a Rossman fold and harbors a dehydrogenase active site capable of binding a nicotinamide adenine dinucleotide cofactor[22]. TRMT10C and SDR5C1 form a stable subcomplex that binds pre-tRNA and is sufficient to catalyze methylation, but not 5′ cleavage of the substrate[7,15]. Cleavage requires PRORP, the catalytic subunit of mtRNase P[7]. It contains an N-terminal pentatricopeptide repeat (PPR) domain, a central zinc-binding domain and a C-terminal NYN (Nedd4-BP1, YacP nuclease) metallonuclease domain, and is thus homologous to single-subunit PRORPs[23]. In contrast to single-subunit PRORPs, however, human PRORP requires TRMT10C and SDR5C1 for pre-tRNA cleavage[7]. Structures of truncated variants of human PRORP demonstrate its resemblance to plant PRORP1, but suggest an autoinhibited conformation in the absence of substrate and TRMT10C or SDR5C1 (refs. [23,24]).

Although individual structures of the subunits have been reported, no high-resolution structural information on the complete mtRNase P complex is available. Therefore, the molecular basis for its dual function and the reasons for the emergence of this unique machinery are not known. In particular, it is not clear how mtRNase P recognizes its substrate, as conserved elements recognized by RNA-based RNase P enzymes and plant PRORP1 are variable or absent in mammalian mitochondrial tRNAs[25–28]. Genetic and biochemical studies have shown that mitochondrial RNA processing proceeds in hierarchical order, with 5′ cleavage preceding 3′ cleavage[5,6,29], and both processes likely occur cotranscriptionally[30,31]. In vitro data suggest that the TRMT10C–SDR5C1 subcomplex may act as a processing platform to facilitate sequential RNA processing steps[29], but the molecular basis for this is unknown.

[1]Department of Cellular Biochemistry, University Medical Center Göttingen, Göttingen, Germany. [2]Research Group Structure and Function of Molecular Machines, Max Planck Institute for Biophysical Chemistry, Göttingen, Germany. [3]Department of Molecular Biology, Max Planck Institute for Biophysical Chemistry, Göttingen, Germany. [4]Cluster of Excellence 'Multiscale Bioimaging: from Molecular Machines to Networks of Excitable Cells' (MBExC), University of Göttingen, Göttingen, Germany. ✉e-mail: hauke.hillen@med.uni-goettingen.de

To investigate the mechanism of mitochondrial pre-tRNA processing and methylation, we reconstituted human mtRNase P and determined its structure using single-particle cryo-EM at an overall resolution of 3.0 Å. The structure contains TRMT10C, four copies of SDR5C1, and PRORP in its active conformation. The substrate pre-tRNA is tightly bound, with the methyl-acceptor base 9 flipped into the TRMT10C methyltransferase domain and the 5′ end positioned in the PRORP active site for cleavage. The structure reveals the architecture of human mtRNase P, including previously unobserved parts of TRMT10C and PRORP that interact extensively with the RNA substrate and with each other. These structural insights provide the molecular basis for recognition, cleavage and methylation of mitochondrial pre-tRNAs and rationalize the unique emergence of a multisubunit proteinaceous RNase P in mammalian mitochondria.

## Results

**Structure of human mitochondrial RNase P.** To reconstitute human mtRNase P, we recombinantly expressed and purified the TRMT10C–SDR5C1 subcomplex and PRORP (Extended Data Fig. 1a). Upon incubation of all three subunits with an in vitro-transcribed pre-tRNA comprising the human mitochondrial tRNA$^{Tyr}$ and flanking sequences (pre-tRNA$^{Tyr}$), we observed cleavage of the 5′ leader, demonstrating that the recombinant complex is functional (Extended Data Fig. 1b). Assembly of the human mtRNase P complex in Mg$^{2+}$-depleted conditions allowed us to obtain a stable complex suitable for structural analysis (Extended Data Fig. 1c,d). We determined the structure of human mtRNase P by single-particle cryo-EM at an overall resolution of 3.0 Å (Table 1 and Extended Data Fig. 2). The resulting density maps allowed us to fit previous crystal structures of TRMT10C, SDR5C1 and PRORP, to extend the structural models substantially, and to model the pre-tRNA$^{Tyr}$ substrate, resulting in a complete structure of the mtRNase P complex (Fig. 1 and Methods).

Human mtRNase P adopts an elongated structure around the centrally embedded pre-tRNA (Fig. 1a,c,d). The base of the complex is formed by the SDR5C1 tetramer. The C-terminal methyltransferase domain of TRMT10C resides on the face of this base, anchored by an 'adapter helix' that connects it to the NTD. The RNA adopts the typical L-shaped tRNA fold and engages in extensive contacts with TRMT10C and PRORP (Fig. 1b,c,d). TRMT10C encases the tRNA between its methyltransferase domain and the previously unresolved NTD (Fig. 1d). PRORP binds on top of the RNA substrate and contacts TRMT10C via two interfaces formed by the PPR and nuclease domains, respectively. Both the 5′ and the 3′ ends of the pre-tRNA are located at the distal end of the complex, and the PRORP nuclease domain is positioned precisely above the 5′ cleavage site.

**The TRMT10C–SDR5C1 subcomplex.** The mtRNase P structure shows how TRMT10C interacts with SDR5C1 to form a stable TRMT10C–SDR5C1 subcomplex (Fig. 2a). SDR5C1 forms a rectangular, symmetric tetramer, as previously reported[22]. Its dehydrogenase active sites are near the binding interface with TRMT10C, and each SDR5C1 monomer has a NADH cofactor bound in its dehydrogenase active site (Extended Data Fig. 3a). TRMT10C interacts with the SDR5C1 tetramer through protein-protein interactions, as predicted[7,15,19,29]. The C-terminal methyltransferase domain of TRMT10C binds on top of the SDR5C1 tetramer, with its active site facing sideways (Fig. 2a). It is anchored to the SDR5C1 tetramer by the adapter helix (TRMT10C α4, residues 183–202), which precedes the methyltransferase domain and was not included in the previous crystal structure[19]. This helix inserts into a symmetrical hydrophobic binding groove on the face of the tetramer formed by all four SDR5C1 monomers (Fig. 2b). N-terminal of the adapter helix, residues 175–182 of TRMT10C form an 'adapter loop', which

is stabilized by hydrophobic interactions with SDR5C1. The previously unresolved NTD of TRMT10C consists of a two-helix bundle (α1–2) and a long 'connector helix' (α3), which connects it to the adapter loop (Figs. 1d and 2a,d).

Due to its symmetry, the SDR5C1 base has a second identical binding groove on the opposite side, suggesting that two TRMT10C monomers could bind to the SDR5C1 tetramer in two different orientations related by a 180° rotation, which would be chemically equivalent. Consistent with this, we observed diffuse density on the

## Table 1 | Cryo-EM data collection, refinement and validation statistics

| | Map 1 global (EMD-13002, PDB 7ONU) | Map 2 PRORP focused (EMD-13002, PDB 7ONU) |
|---|---|---|
| **Data collection and processing** | | |
| Magnification | ×105,000 | ×105,000 |
| Voltage (kV) | 300 | 300 |
| Electron exposure (e⁻/Å²) | 39.6 | 39.6 |
| Defocus range (μm) | 0.2–2.5 | 0.2–2.5 |
| Pixel size (Å) | 0.834 | 0.834 |
| Symmetry imposed | C1 | C1 |
| Initial particle images (no.) | 6,150,677 | 6,150,677 |
| Final particle images (no.) | 88,081 | 88,081 |
| Map resolution (Å) | 3.0 | 3.1 |
| FSC threshold | 0.143 | 0.143 |
| Map resolution range (Å) | 4.7–2.7 | 4.7–3.0 |
| Map sharpening B factors (Å²) | −61.4 | −72.3 |
| **Refinement** | | |
| Cryo-EM map used | Composite map from map 1 and 2 | |
| Initial models used (PDB codes) | 1U7T, 5NFJ, 4XGL, 3TUP | |
| Model resolution (Å) | 3.0 | |
| FSC threshold | 0.5 | |
| Model composition | | |
| Nonhydrogen atoms | 15,088 | |
| Protein/nucleotide residues | 1,764/66 | |
| Ligands | ZN:1, MG:1, NAI:4 | |
| B factors (Å²), min./max./mean | | |
| Protein | 22.58/70.49/36.85 | |
| Nucleotide | 38.29/70.72/48.75 | |
| Ligand | 33.68/49.62/37.59 | |
| R.m.s. deviations | | |
| Bond lengths (Å) | 0.004 | |
| Bond angles (°) | 0.941 | |
| Validation | | |
| MolProbity score | 1.30 | |
| Clashscore | 3.39 | |
| Rotamer outliers (%) | 0.07 | |
| Ramachandran plot | | |
| Favored (%) | 97.60 | |
| Allowed (%) | 2.40 | |
| Disallowed (%) | 0.00 | |

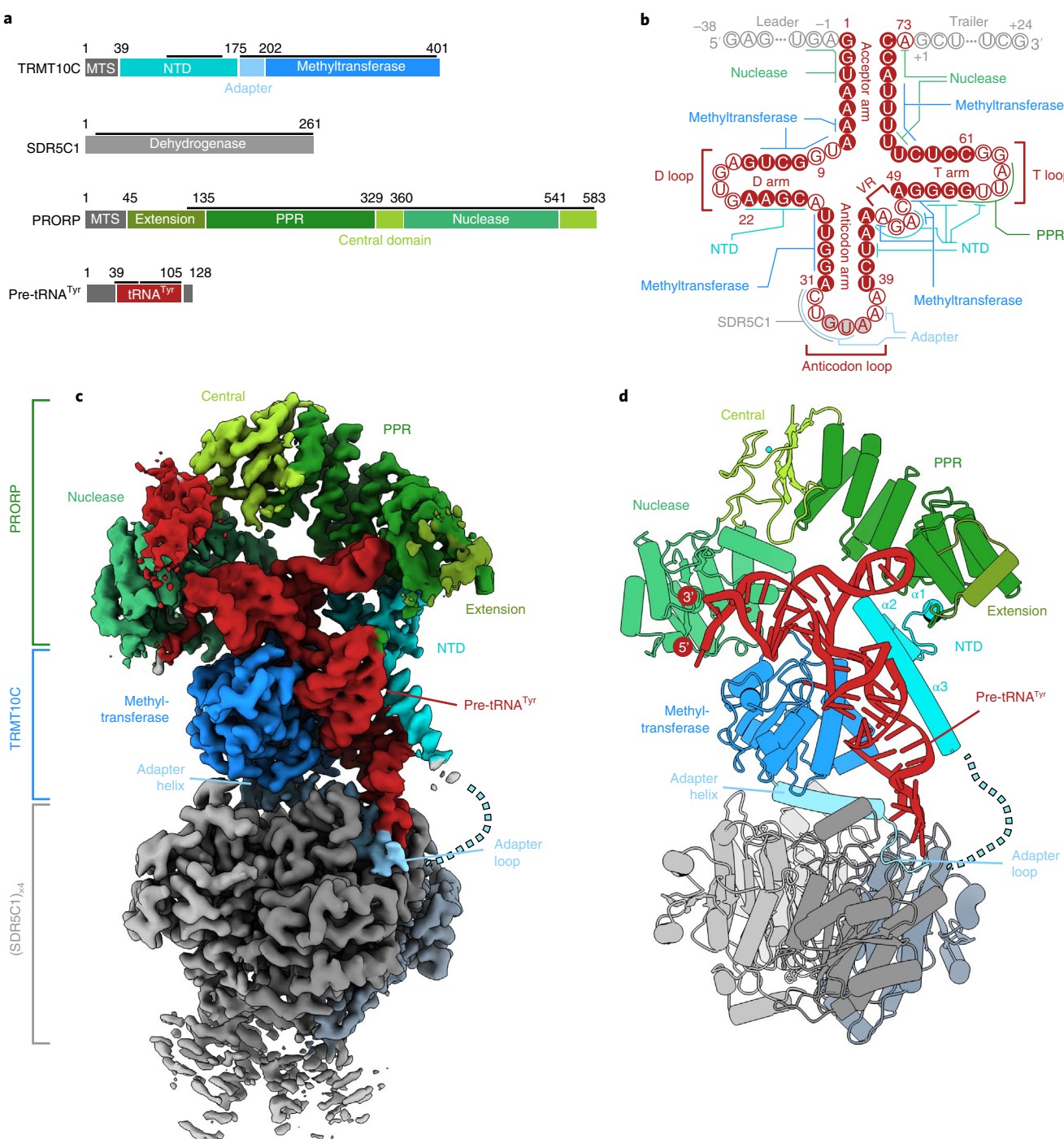

**Fig. 1 | Structure of the human mitochondrial RNase P complex. a**, Domain representation of mtRNase P subunits and substrate pre-tRNA[Tyr]. TRMT10C domains are shown in shades of blue; the four SDR5C1 subunits are shown in shades of gray; PRORP domains are shown in shades of green and the pre-tRNA is shown in red. Protein and RNA regions modeled in the structure are indicated with black lines. This coloring is used throughout. **b**, Schematic representation of the pre-tRNA[Tyr] with structural elements of the tRNA and RNA-protein interactions in the mtRNase P complex indicated. The tRNA[Tyr] sequence is shown in red and the leader and trailer sequences are shown in gray. The variable region is labeled as 'VR'. Base-paired nucleotides are shown as solid circles and positioned in parallel. Unpaired nucleotides are shown as hollow circles. The three anticodon nucleotides are labeled and shown as gray circles with red outline. The numbering is according to canonical tRNA numbering[33] and is used throughout (see Supplementary Note 1 for cross-referencing of canonical nucleotide positions in tRNA[Tyr] to the numbering in the deposited coordinates file). Leader sequences at the 5′ end of tRNA are labeled −38 to −1, and trailer sequences at the 3′ end are labeled +1 to +24. **c**, Cryo-EM density map of the mtRNase P complex. Coloring as in **a**. **d**, Cartoon representation of the structure of the mtRNase P complex. Helices α1–3 of the previously unmodeled TRMT10C NTD are labeled.

other side of the SDR5C1 tetramer in some two-dimensional (2D) class averages (Extended Data Fig. 2b) and in the three-dimensional (3D) reconstruction of mtRNase P (Fig. 1c). Further focused classification revealed two particle subpopulations with approximately

equal distribution, which, upon independent refinement, indicate a second TRMT10C copy and tRNA bound either in parallel or antiparallel orientation (Extended Data Fig. 3b,c). Thus, the TRMT10C–SDR5C1 subcomplex may bind two substrate RNAs simultaneously,

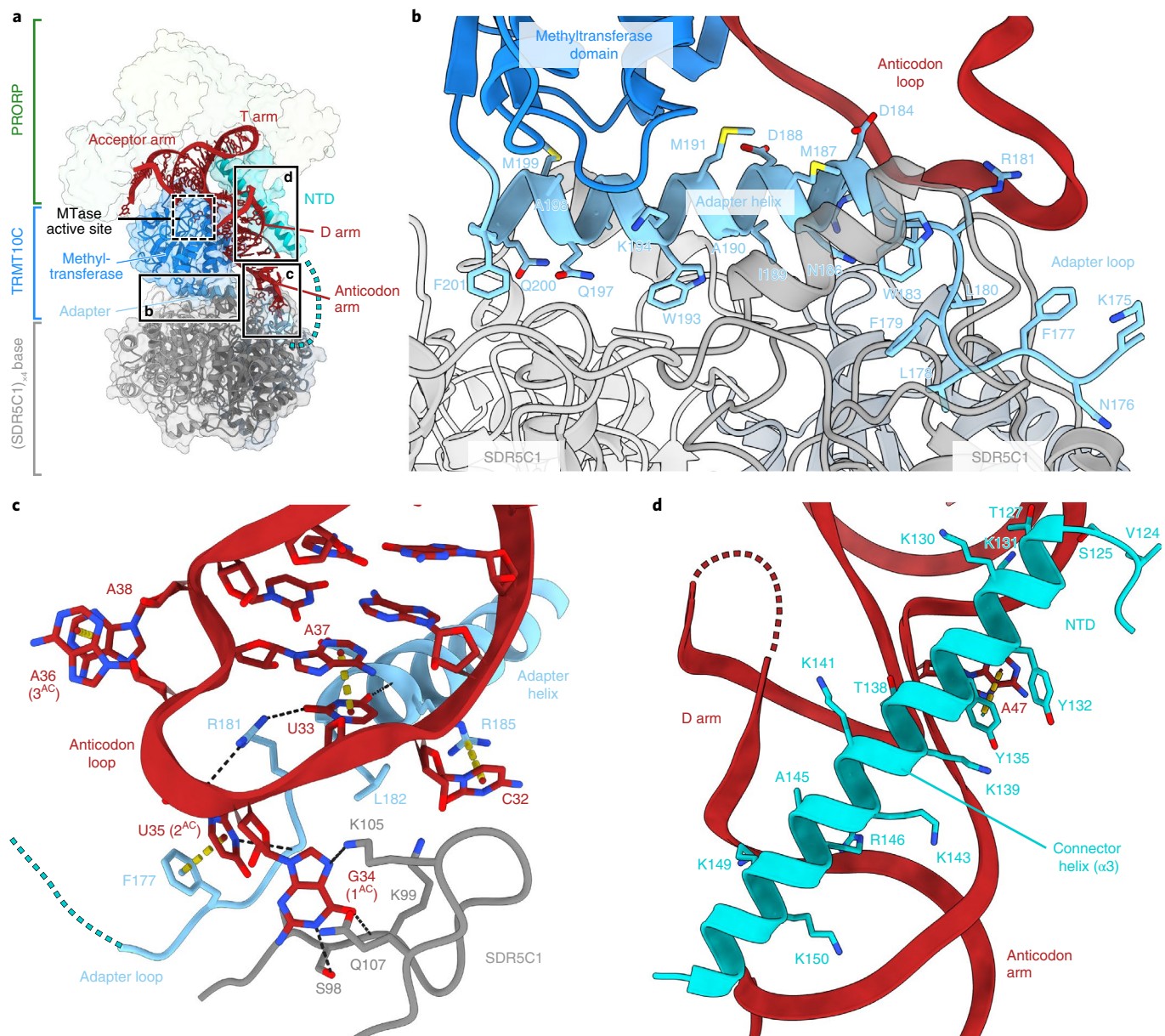

**Fig. 2 | Structure and RNA interactions of the TRMT10C–SDR5C1 subcomplex. a**, Overview of the TRMT10C–SDR5C1 subcomplex with the regions enlarged in **b**, **c** and **d** indicated. TRMT10C, SDR5C1 and pre-tRNA$^{Tyr}$ are shown as ribbons, with TRMT10C, SDR5C1 and PRORP additionally shown as transparent surfaces. MTase, methyltransferase. **b**, Close-up view of the interaction between the SDR5C1 tetramer and TRMT10C. Residue side chains in the TRMT10C adapter helix and adapter loop are shown as sticks. **c**, Close-up view of the interaction of the TRMT10C–SDR5C1 subcomplex with the pre-tRNA$^{Tyr}$ anticodon loop. Residues within 4 Å of the anticodon loop are shown as sticks. The phosphate backbone of the tRNA is shown as a ribbon and the bases are shown as sticks. Anticodon bases are labeled as 1–3$^{AC}$. Stacking interactions within 4.2 Å are shown as yellow dashed cylinders and probable hydrogen bonds are shown as black dashed lines (see Methods for details). These representations are used throughout. **d**, Interaction of the TRMT10C connector helix with the pre-tRNA$^{Tyr}$. The connector helix of TRMT10C is positioned between the D arm and anticodon arm of the tRNA. TRMT10C residues within 4 Å of the pre-tRNA$^{Tyr}$ are shown as sticks. A stacking interaction between A47 of pre-tRNA$^{Tyr}$ and Y135 of TRMT10C is shown.

consistent with a previously proposed 2:4 (TRMT10C:SDR5C1) stochiometry[7,19,32].

**Substrate binding by the TRMT10C–SDR5C1 subcomplex.** The structure reveals how the TRMT10C–SDR5C1 subcomplex binds the pre-tRNA substrate. The complex forms a shape-complementary platform for the canonical L-shape of tRNAs (Fig. 2a). The acceptor arm of the pre-tRNA resides on top of the TRMT10C methyltransferase, while the anticodon arm lies adjacent and reaches down to the SDR5C1 tetramer. The tip of the anticodon arm runs along the tetramer near a lysine-rich loop in SDR5C1 (residues

97–107), which may form multiple interactions with G34 (canonical tRNA numbering[33], see also Methods and Supplementary Note 1) through its backbone and side chains of S98, K105 and, possibly, Q107 (Fig. 2c). These interactions are probably not base specific, as this position is not conserved in mitochondrial tRNAs. The anticodon loop is positioned directly above the adapter loop and helix of TRMT10C, which form extensive interactions with the RNA. In particular, F177 stacks against U35, the second base of the anticodon triplet, and R185 stacks against C32. R181 protrudes into the anticodon loop and interacts with the C2 carbonyl of U33 (Fig. 2c). This interaction is specific to pyrimidine nucleobases and

is likely crucial for correct substrate recognition, as this position is conserved as U or C in tRNAs, and mutations of R181 cause mitochondrial RNA processing defects and respiratory chain deficiency in patients (Extended Data Fig. 4)[34].

The pre-tRNA additionally forms multiple interactions with the methyltransferase and NTD of TRMT10C, which encase the anticodon arm (Fig. 1d). On one side, the methyltransferase domain forms a shape- and charge-complementary surface through a number of positively charged residues (Extended Data Fig. 5a). On the other side, the connector helix of the NTD runs along a groove in the tRNA in between the D arm, the anticodon arm and the T arm. The RNA-facing side of this helix is also lined with basic residues to accommodate the negative charges of the RNA (Fig. 2d). In addition, Y135 stacks against A47 in the variable region. These interactions collectively appear to distort the structure of the tRNA, as the groove between the D arm and the anticodon arm is widened and the anticodon loop is displaced compared to the canonical tRNA structure (Fig. 3).

Previous studies of archaeal and yeast TRM10 enzymes suggest that the NTD of these enzymes is also involved in tRNA substrate binding[35–37]. However, structural comparison shows that the TRMT10C NTD adopts an unrelated fold and the elements that interact with the tRNA are not conserved (Extended Data Fig. 6 and Supplementary Note 2), suggesting that the substrate recognition mechanism differs between TRMT10C and other TRM10 enzymes. This may also explain the requirement of SDR5C1 for efficient tRNA methylation[15], because the anticodon binding elements of TRMT10C are stabilized by its interaction with SDR5C1.

In summary, the TRMT10C–SDR5C1 subcomplex contacts all four arms of the pre-tRNA and interacts with the substrate through both nonspecific and specific interactions.

**Methylation of mitochondrial pre-tRNAs.** The interactions between TRMT10C, SDR5C1 and the pre-tRNA explain how the conserved methylation of mitochondrial tRNAs at position 9 is established. The methyltransferase domain of TRMT10C is located next to the pre-tRNA substrate and contacts the acceptor arm, the D arm and the anticodon arm (Figs. 1 and 2a). The substrate is positioned and held in place by its interactions with the TRMT10C NTD and SDR5C1, explaining why both are required for methylation[15,19]. The methyltransferase active site is positioned directly adjacent to the junction between the acceptor arm and the D arm, where G9 is flipped out of the tRNA fold and buried into the active site, stacking against the conserved residue V313 (Fig. 4b). On one side of the binding pocket, N222 may interact with the 2′-OH of the ribose, and the invariant residue Q226, which is essential for methylation[19,37], may interact with N3 and the primary amine (exocyclic N6) of the bound base, potentially ensuring selectivity for purines over pyrimidines. On the opposite side, N350 and N348 reach towards the substrate and may form interactions with the carbonyl oxygen of the guanine base. Modeling of an adenine base instead of the guanine observed in our structure indicates that the active site can accommodate both purine bases, consistent with the dual specificity of TRMT10C and the occurrence of either G or A at position 9 in mitochondrial tRNAs (Extended Data Fig. 5d).

Comparison with the previously reported SAM-bound crystal structure of the TRMT10C methyltransferase domain[19] provides a model for the precatalytic state immediately before methyl transfer (Fig. 4c and Extended Data Fig. 5d). Superposition shows that the structure of the methyltransferase domain remains unchanged upon RNA binding (all-atom root mean squared deviation (r.m.s.d.) = 0.56 Å) and places the methyl group of the SAM cofactor in the immediate vicinity (<2.5 Å) of N1 of the substrate base. The only major difference between the SAM-bound and tRNA-bound structures is the conformation of the conserved loop motif II

(residues 314–319), which interacts with the SAM cofactor via D314 in the crystal structure. In the SAM-free mtRNase P complex, this loop adopts a different conformation (Extended Data Fig. 5b,c). However, the conformation observed in the presence of SAM would be possible with bound RNA substrate without major clashes, and it may thus rearrange upon cofactor binding, as has been shown for other TRM10 enzymes[35–37].

In conclusion, these data provide the structural basis for N1-methylation at position 9 of mitochondrial tRNAs.

**Recruitment and activation of the nuclease subunit PRORP.** The structure of the mtRNase P complex reveals the active conformation of PRORP with the substrate positioned for 5′ cleavage. PRORP forms an arch-like structure atop the TRMT10C–SDR5C1–pre-tRNA complex, interacting with both the pre-tRNA and TRMT10C (Fig. 1d). In previous crystal structures of human PRORP, the protein adopted 'closed' conformations in which active site residues were sequestered by intramolecular interactions between the nuclease and central domains[23,24]. In the mtRNase P complex, PRORP adopts a more 'open' conformation with respect to the angle between its nuclease and PPR domains (Fig. 5a). Compared to the previously observed states, the nuclease domain is rotated outwards by ~25° to allow accommodation of the pre-tRNA substrate, which would otherwise lead to clashes (Fig. 5a and Extended Data Fig. 7a). The open conformation of PRORP is stabilized by interactions with the TRMT10C–SDR5C1 complex and with the RNA.

On one side of PRORP, the PPR domain contacts the T arm of the pre-tRNA substrate and the NTD of TRMT10C (Fig. 5b,c). The interaction with TRMT10C is mediated by loops between helices α4–α7 of the PRORP PPR domain (residues 174–176, 205–209 and 238–241), which contact α1 and α3 of TRMT10C (Fig. 5b), explaining why the N-terminal region of PRORP is critical for activity[23]. The curved arrangement of PPR repeats, together with α3 of TRMT10C, creates a positively charged groove that accommodates the T arm of the substrate through backbone stabilization (Figs. 5c and 2d). In addition, the conserved residue Y183 stacks against the ribose of A57 (Fig. 5c). These interactions do not appear to be sequence specific and hence do not correspond to the previously proposed 'PPR code' observed in plant PPR proteins[38], as also suggested by previous studies of single-subunit PRORP enzymes[27,39,40].

On the opposite side of PRORP, the nuclease domain contacts the methyltransferase domain of TRMT10C. This is mediated by a loop between α12 and α13 of PRORP (residues 381–387), which contacts α4 of the TRMT10C methyltransferase (numbering based on methyltransferase domain structure[19]) (Fig. 5d). The acceptor arm and leader nucleotides of the pre-tRNA substrate are contacted by basic residues in the nuclease domain that line the path of the RNA backbone (R445, K415 and R502) (Fig. 5e), positioning the scissile phosphodiester bond in the nuclease active site. Thus, interactions with TRMT10C and pre-tRNA at two interfaces, via the PPR and nuclease domains, stabilize the open conformation of PRORP and position the substrate for precise cleavage.

The open conformation of PRORP leads to release of the intramolecular interactions in the active site observed in the previous crystal structures. First, the rotation of the nuclease domain moves the active site away from the lariat loop (residues 560–575), an element in the central domain that sequestered active site residue D499 in the previously observed state (Extended Data Fig. 7b)[23]. The loop containing D499 is also rearranged, which brings this residue closer to the other three catalytic aspartates D409, D478 and D479. Second, D478, which was sequestered by interaction with R445 in one of the previous structures[24], becomes available for catalysis, as R445 is repositioned to stabilize the phosphate group of the −1 nucleotide (Fig. 5e). Third, R498 is rotated away from the active site, where it occupied one of the proposed catalytic metal-binding sites in a previous PRORP structure[23].

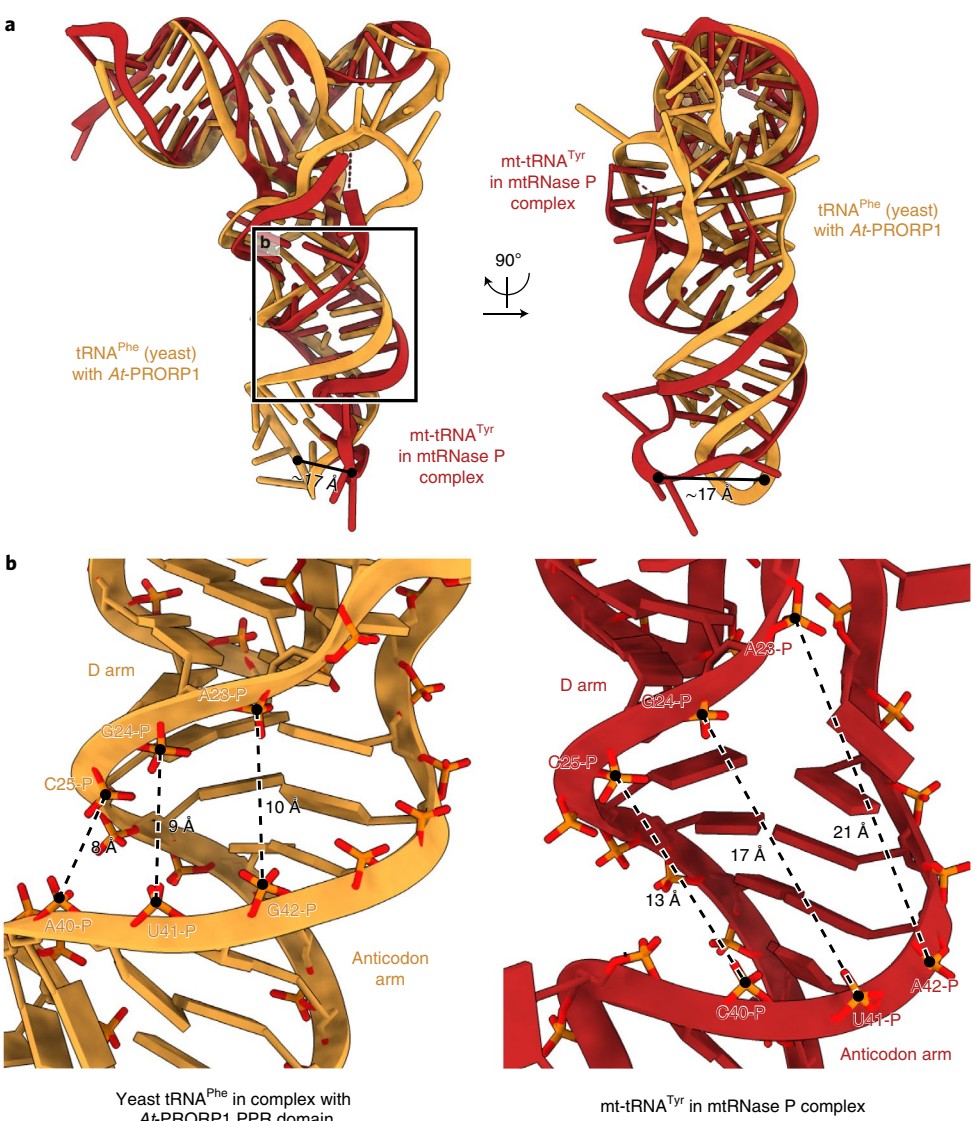

**Fig. 3 | The pre-tRNA adopts a distorted structure. a,** Comparison of tRNA structures for human pre-(mt)tRNA<sup>Tyr</sup> in the mtRNase P complex (red) and yeast tRNA<sup>Phe</sup> in complex with the *A. thaliana* PRORP1 (*At*-PRORP1) PPR domain (orange) (PDB 6LVR)[27]. The two tRNA structures are aligned with the acceptor and T arms (pre-tRNA<sup>Tyr</sup>: residues 1–7 and 44–65; tRNA<sup>Phe</sup>: residues 1–7 and 49–72). In the mtRNase P complex, the pre-tRNA<sup>Tyr</sup> adopts a distorted structure compared to a canonical tRNA, resulting in an approximately 17 Å displacement of the anticodon loop (distance between phosphate atoms between anticodon residues 1 and 2, shown as black circles). **b,** Distance comparison between the D arm and the anticodon arms of pre-tRNA<sup>Tyr</sup> in the mtRNase P complex and tRNA<sup>Phe</sup> bound to the *At*-PRORP1 PPR domain. Distances between phosphorous atoms of nucleotide pairs 25–40, 24–41 and 23–42 (canonical tRNA numbering[33]) in both tRNAs are shown as dashed lines. The distance between the D and anticodon arms of pre-tRNA<sup>Tyr</sup> in the mtRNase P complex (right) is considerably larger than in a tRNA bound to *At*-PRORP1 (left).

The resulting active site arrangement is highly similar to that of *Arabidopsis thaliana* PRORP1 (*At*-PRORP1), which is presumed to adopt an active state even without bound substrate (Extended Data Fig. 7c)[41]. In between the scissile phosphate and the catalytic residue D499, we observe a density consistent with the location of one of the two catalytic metals (metal 2) in the *At*-PRORP1 crystal structure (Extended Data Fig. 7c). The structure therefore represents the active conformation of PRORP primed for cleavage, in which catalytic residues are arranged to coordinate two metal ions required for endonucleolytic cleavage and the scissile phosphodiester bond is positioned in the active site (Fig. 5f).

Taken together, the structure of mtRNase P reveals the active conformation of PRORP and shows how this conformation is stabilized by interactions between the protein subunits and the substrate.

**Comparison to RNase P ribozymes and single-subunit PRORPs.**
The structure of mtRNase P shows that the mechanism by which it recognizes and positions its substrate differs strongly from those used by ribozyme-based RNase P and single-subunit PRORPs. Both bacterial and eukaryotic RNase P interact with the 'elbow' region of the pre-tRNA formed by the T and D loops and with the acceptor arm, while the anticodon arm faces away from the enzymes. The substrate is positioned through interactions of the catalytic RNA with the conserved G/U19-C56 base pair formed between the TΨC and D loops, and through interactions of the 5′ leader with accessory proteins (Fig. 5a,b)[25,26]. Although structurally unrelated, the single-subunit *At*-PRORP1 also interacts with the G/U19-C56 base pair of the pre-tRNA substrate[27,42]. Its PPR domain provides a positively charged binding groove and forms base-stacking interactions with C56 and base-specific interactions with G19 (Fig. 6a,c).

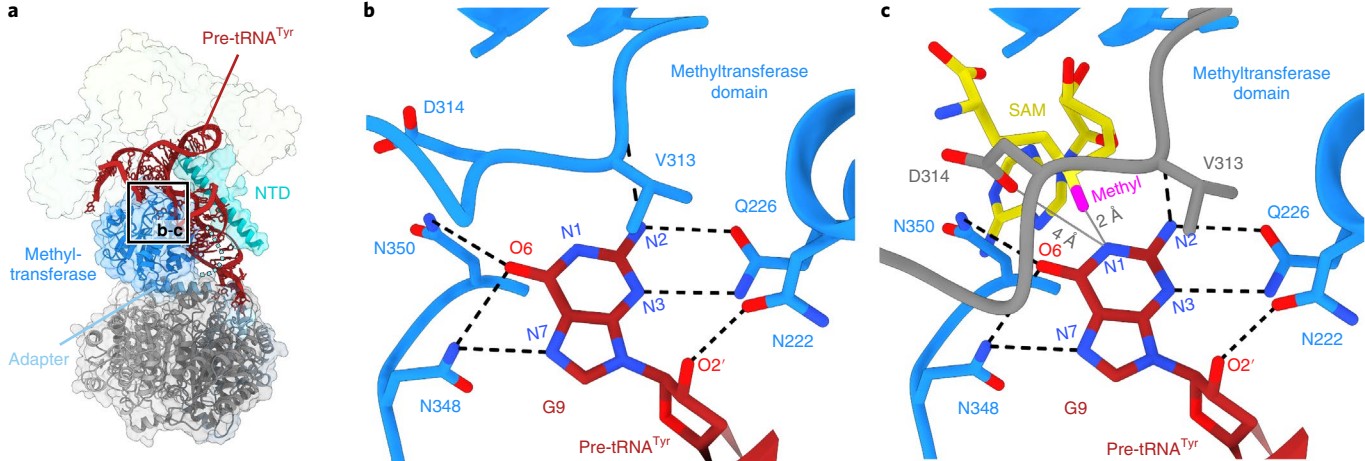

**Fig. 4 | The TRMT10C methyltransferase active site. a**, Overview of the TRMT10C–SDR5C1 subcomplex with the region enlarged in **b** and **c** indicated. TRMT10C, SDR5C1 and pre-tRNA$^{Tyr}$ are shown as ribbons, with TRMT10C, SDR5C1 and PRORP additionally shown as transparent surfaces. **b**, Close-up view of the methyltransferase active site of TRMT10C. Nucleotide G9 of the pre-tRNA$^{Tyr}$ and active site residues within 4 Å of G9 and D314 of TRMT10C are shown in stick representation. Potential hydrogen bonds are shown as dashed black lines. Methyl-acceptor atom N1 and additional atoms of G9 potentially involved in active site interactions are labeled. **c**, Precatalytic model of the TRMT10C methyltransferase active site for G9 methylation. To generate this model, the crystal structure of the substrate-free SAM-bound TRMT10C methyltransferase domain (PDB 5NFJ)[19] was superimposed with the TRMT10C methyltransferase domain in mtRNase P. The model is composed of the active site of substrate-bound TRMT10C in the mtRNase P complex, shown as in **b**, with the location of the SAM cofactor and the conformation of residues 310 to 320 (motif II loop) of TRMT10C modeled as in the SAM-bound crystal structure. SAM is shown in yellow with the methyl group in magenta, and the motif II loop derived from PDB 5NFJ is shown in gray.

While the G/U19-C56 base pair is highly conserved in bacterial, cytosolic and plant tRNAs, mammalian mitochondrial tRNAs have highly variable T and D loops and generally lack this element[28]. Consequently, human mtRNase P does not form specific interactions with the elbow, and its RNA interactions differ from those observed in *At*-PRORP1. Although the positively charged groove and the residue that forms a stacking interaction with the C56 nucleobase in *At*-PRORP1 are conserved in human PRORP (Y140 in *At*-PRORP1, Y183 in human PRORP), the residue that interacts with G19 (Y133 in *At*-PRORP1) is not (Fig. 6a,c,d and Supplementary Note 3). Y183 of human PRORP is also not involved in base-stacking interactions with pre-tRNA$^{Tyr}$ and instead interacts with the backbone ribose of A57 (Figs. 5c and 6d). While the tip of the D loop (U19–G21) is in close proximity to the N-terminal region of PRORP (residues 135–138), both RNA and protein are poorly resolved in these regions, arguing against specific interactions. However, additional contacts may be formed with other mitochondrial tRNAs, as some contain longer D and T loops. Thus, the mtRNase P complex can likely also accommodate more canonical tRNA substrates, consistent with observations that it can also cleave cytosolic or bacterial tRNAs in vitro[15,43].

The lack of specific interactions by PRORP is contrasted by the intricate interactions between the pre-tRNA and the TRMT10C–SDR5C1 subcomplex, which recognizes several features of mitochondrial pre-tRNAs. First, it forms a shape- and charge-complementary binding platform that encases the pre-tRNA between the TRMT10C methyltransferase and N-terminal domains. Second, the TRMT10C connector helix probes the tRNA structure and/or stabilizes its distorted conformation. Third, TRMT10C specifically interacts with base 33 in the anticodon loop, which is conserved as pyrimidine. Finally, binding of the substrate base to the TRMT10C methyltransferase may further stabilize the complex. Thus, the mitochondria-specific subunits TRMT10C and SDR5C1 appear to compensate for the absence of structural elements in mitochondrial tRNAs that are recognized by other RNase P enzymes, as has been suggested[10].

## Discussion

Here we present the structure of human mtRNase P, which carries out the initial steps of mitochondrial RNA processing to produce individual RNA species from primary transcripts. The structure reveals how pre-tRNA is recognized, cleaved and methylated and suggests an explanation for the emergence of the evolutionarily unique multimeric proteinaceous mtRNase P.

The structure shows that mtRNase P employs a unique mechanism of substrate recognition. While other RNase P enzymes recognize pre-tRNAs through specific interactions with conserved elements in the elbow region, these elements are variable among mitochondrial tRNAs. Therefore, mtRNase P recognizes other general structural features as well as conserved elements in mitochondrial tRNAs through its mitochondria-specific subunits TRMT10C and SDR5C1. This suggests that the multisubunit mtRNase P has co-evolved with the variability in mammalian mitochondrial tRNAs, although it is not clear whether it has emerged to compensate for this variability or enabled it in the first place. Both TRMT10C and SDR5C1 are dual function, and the structure of mtRNase P shows how their functionally unrelated folds, such as the SDR5C1 dehydrogenase tetramer, are repurposed for a role in pre-tRNA processing. Such structural repurposing has also been observed for the mitochondrial transcription factor TF2BM[44], and may be a common theme in mitochondrial gene expression.

The structure of mtRNase P also provides a possible model for the activation of human PRORP. Previous crystal structures of PRORP revealed a closed conformation incompatible with substrate binding and cleavage, which led to the suggestion that apo-PRORP adopts an autoinhibited state[23,24]. However, the constructs used lacked parts of the PPR domain and were inactive in vitro, and it has thus been debated whether these structures represent physiological states[10,11]. In the mtRNase P complex, PRORP adopts an open, active conformation, which is stabilized by interactions with TRMT10C–SDR5C1 and the pre-tRNA. If apo-PRORP adopts the closed state, the enzyme must therefore undergo rearrangements upon binding to the TRMT10C–SDR5C1–pre-tRNA complex, as has been suggested[23,24]. Although it remains tentative without a full-length

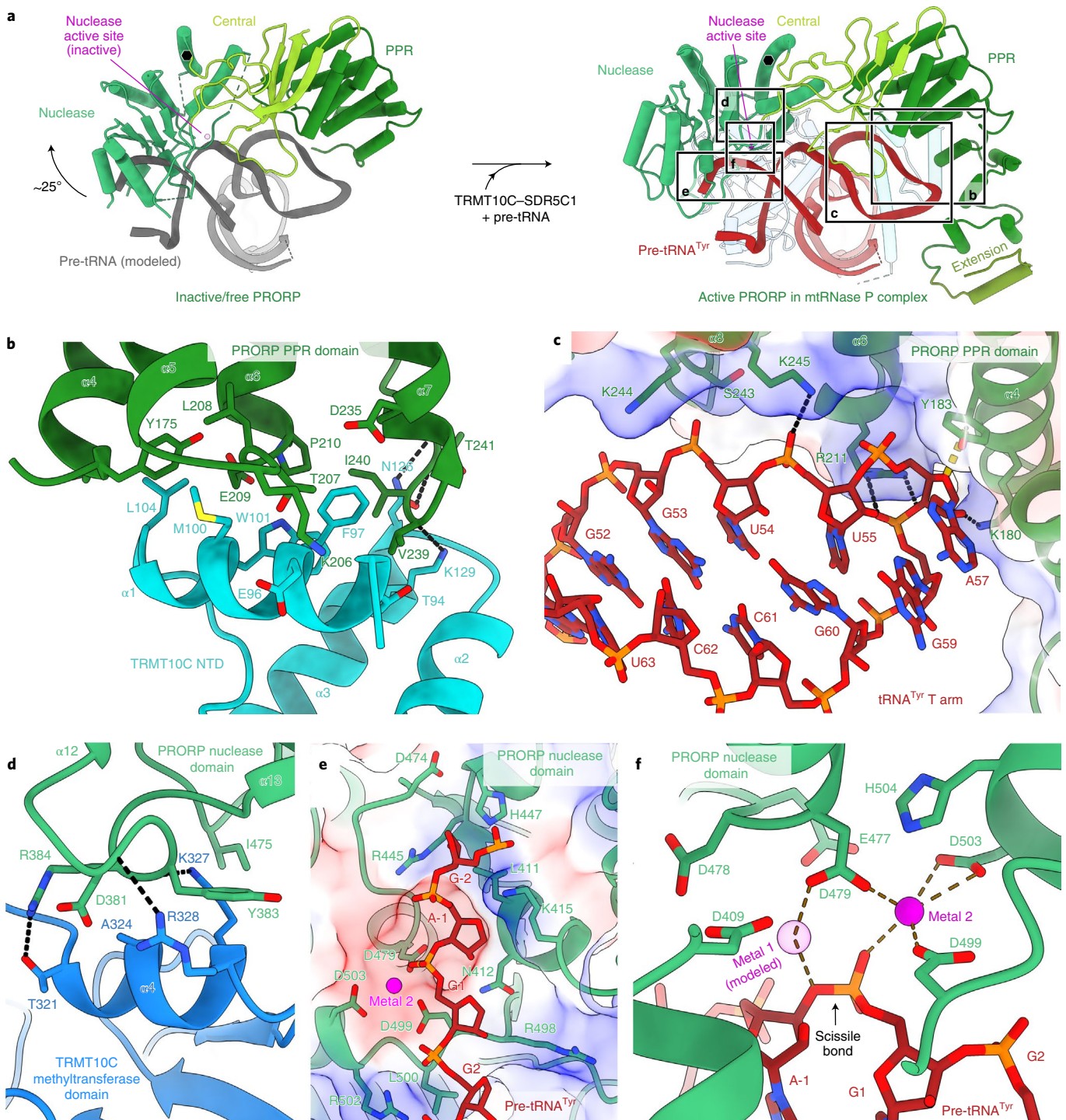

**Fig. 5 | Structure and substrate interactions of active human PRORP. a**, Comparison of free/inactive human PRORP (left) (PDB 4XGL)[23] and activated human PRORP in mtRNase P complex (right), aligned via their PPR domains (residues 207–329). Active sites are marked as magenta spheres. In the active conformation, the nuclease domain is rotated outward by approximately 25° to accommodate the pre-tRNA[Tyr]. The axis of rotation is indicated by a black hexagon. Regions shown in panels **b**–**f** are indicated. **b**, Interactions between the NTD of TRMT10C and the PPR domain of PRORP. Residues in the NTD and PPR domain within 4 Å of the interaction interface are shown as sticks. **c**, Interaction of the pre-tRNA[Tyr] T arm with the PPR domain of PRORP. PRORP residues within 4 Å of tRNA are shown as sticks. The surface electrostatic potential of the PPR domain is shown transparently as a three-color gradient scheme from −5 to +5 kcal mol[−1] e[−1] (red, negative; white, neutral; blue, positive). **d**, Interactions between the methyltransferase domain of TRMT10C and the nuclease domain of PRORP. Residues in the methyltransferase domain and nuclease domain within 4 Å of the interaction interface are shown as sticks. **e**, Interactions between the pre-tRNA[Tyr] backbone and the nuclease domain. RNA backbone and PRORP residues within 4 Å are shown as sticks. The surface electrostatic potential of the nuclease domain is shown as in **c** and the active site metal 2 is shown as a solid magenta sphere. **f**, Substrate-bound PRORP active site. Residues in the active site and the pre-tRNA[Tyr] are shown in stick representation. Metal 2 is shown as a solid magenta sphere. Metal 1 was modeled on the basis of the At-PRORP1 crystal structure (PDB 4G24)[41], and is shown as a transparent magenta sphere. Potential coordination interactions with metal 1 and 2 are shown as brown dashed lines.

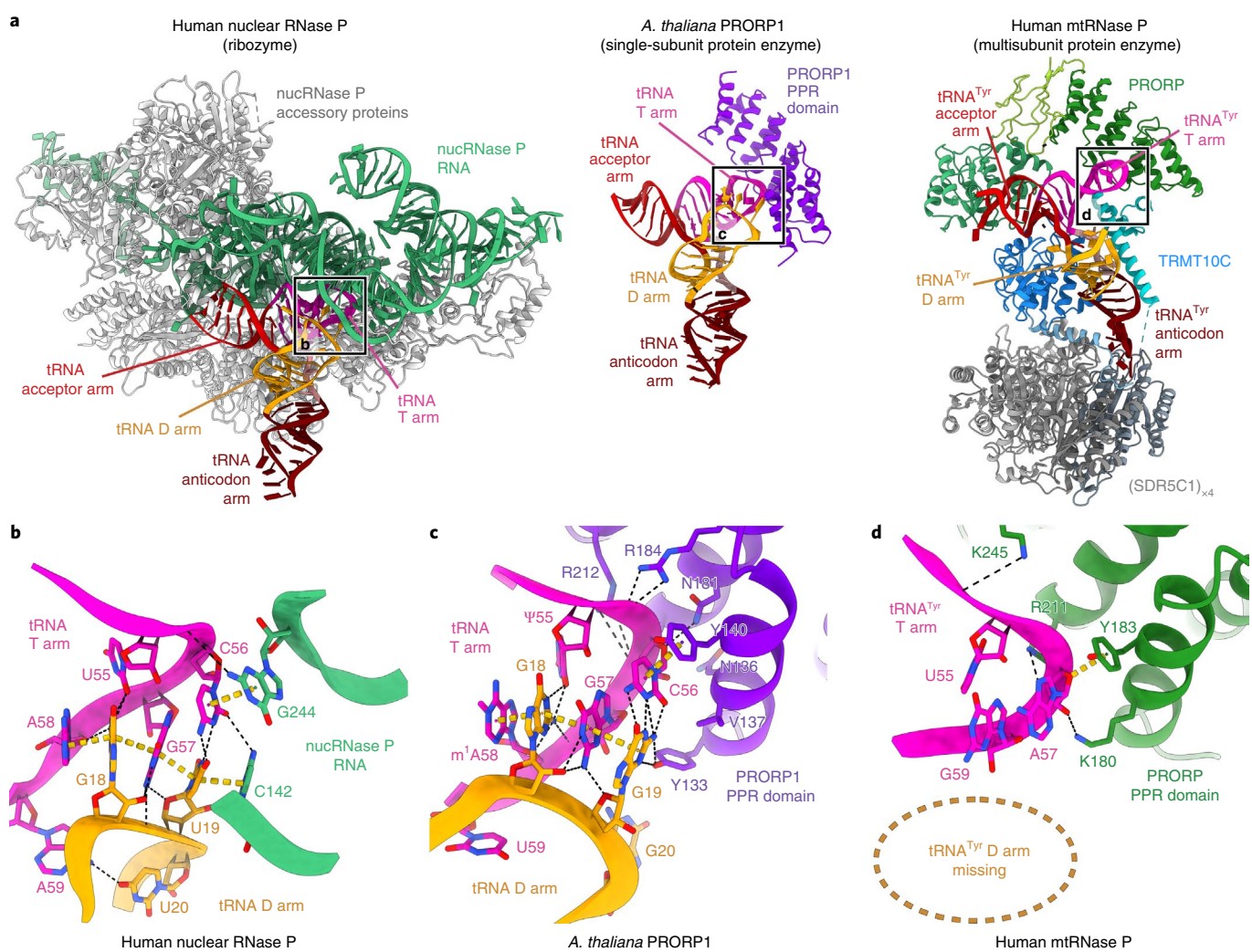

**Fig. 6 | Comparison of substrate recognition by different RNase P enzymes. a**, Structural comparison between human nuclear RNase P ribozyme (PDB 6AHU)[26], *Arabidopsis thaliana* PRORP1 PPR domain in complex with tRNA (PDB 6LVR)[27] and human mtRNAse P in complex with pre-tRNA^Tyr (left to right). Nuclear RNase P (nucRNase P) catalytic RNA and accessory proteins are colored in green and gray, respectively; *At*-PRORP1 PPR domain is colored in purple; and mtRNase P protein domains are colored as in Fig. 1a. The tRNAs are shown colored by structural elements: acceptor arm, red; T arm, pink; D arm, yellow; anticodon loop, dark red; variable region, light brown (not labeled). Elbow regions of tRNAs, enlarged in panels **b**–**d**, are indicated. **b**, Close-up of the tRNA-elbow recognition site in human nuclear RNase P. Bases in the elbow region of the tRNA and RNase P RNA bases within 4 Å are shown as sticks. The conserved interaction between the D and T arm is recognized by the RNA component of RNase P. **c**, Close-up of the tRNA-elbow recognition site in *At*-PRORP1. Bases in the elbow region of tRNA and *At*-PRORP1 residues within 4 Å are shown as sticks. The D and T arm interaction, including the C56-G19 base pair contact, is recognized by the PPR domain of *At*-PRORP1. **d**, Close-up of the tRNA-elbow binding site in the human mtRNase P complex. Bases in the elbow region of pre-tRNA^Tyr and PRORP residues within 4 Å are shown as sticks. Interactions between T and D arms in the elbow region are absent, and PRORP PPR domain residues interact primarily with the RNA backbone.

apo-PRORP structure, the structure of mtRNase P provides a model for how binding to the TRMT10C–SDR5C1–pre-tRNA complex could activate PRORP for cleavage (Supplementary Video 1).

The structure of active human PRORP is the first structure of a PRORP nuclease in complex with substrate RNA and provides insights into the architecture of the active site during catalysis. Both ribozyme-based RNase P enzymes and PRORPs are thought to employ a two-metal-dependent mechanism during which one metal activates the nucleophile while the other stabilizes the transition state[11,45]. However, studies of plant PRORP enzymes suggest that their precise mechanisms differ from those of RNase P ribozymes[46,47]. For *At*-PRORP1, it has been proposed that the active-site metals coordinate the pro-S oxygen of the scissile phosphate, and that D399 acts as a general base to activate a metal-bound water for nucleophilic attack[41]. The active site of substrate-bound human

PRORP is virtually identical to that of *At*-PRORP1 (Extended Data Fig. 7c) and shows that the pro-S oxygen is coordinated by metal 2, and D409 is in an equivalent position to D399 in *At*-PRORP1. Thus, our structural data are consistent with the proposed mechanism of phosphodiester cleavage by PRORPs[41].

The structure of mtRNase P also represents the first structure of a TRM10 methyltransferase in complex with substrate RNA, and shows how mitochondrial pre-tRNAs are methylated at position 9. Previous studies of *Schizosaccharomyces pombe* TRM10 suggested that a conserved aspartate may act as a general base to activate N1 for methylation[37]. However, recent studies on yeast and archaeal enzymes have challenged this view, as single substitutions of the corresponding aspartate did not show a strong effect[36,48]. In TRMT10C, mutation of the conserved active site residues D314 or Q226, to asparagine or alanine, respectively, leads to impaired

methyltransferase activity[15,19]. The structure of mtRNase P shows that Q226 interacts with the substrate base and that D314, which corresponds to the proposed general base in yeast TRM10 (ref. [48]), is in close proximity to the nucleobase (Fig. 3b). In the conformation modeled in the precatalytic state, D314 may thus facilitate deprotonation of the base, which in turn may be stabilized by N350 or N348 (Fig. 3c). Similar contacts may occur when an adenine base is bound in the active site (Extended Data Fig. 5d), and in this case D314 may deprotonate the exocyclic N6 to stabilize the imine tautomer, as was suggested for a bacterial m$^1$A methyltransferase[49]. Thus, the structure of the mtRNase P complex rationalizes TRMT10C active site mutations and is consistent with an important role of D314 in catalysis, as predicted[13].

N1-methylation at position 9 is a modification found in 19 out of 22 human mitochondrial tRNAs[50] and has been proposed to stabilize the mature fold of some of these[51]. We thus hypothesize that the TRMT10C–SDR5C1 complex could act as an RNA chaperone by facilitating or sampling the proper folding of pre-tRNA segments within primary mitochondrial transcripts. The shape- and charge-complementarity of the TRMT10C–SDR5C1 complex may facilitate initial formation and transient binding of cloverleaf-like structures in the RNA. Properly folded substrates may then become tightly bound by additional specific interactions with the anticodon loop and by binding of the TRMT10C NTD, which may undergo transient movements to encase the RNA. Once a properly folded pre-tRNA is bound, methylation can occur and lead to stabilization of the mature cloverleaf structure.

Together with previous data, the structural data presented here lead us to suggest a model for early mitochondrial RNA processing, which we have summarized in a molecular movie (Supplementary Video 1). First, TRMT10C and SDR5C1 form a subcomplex, which can bind and methylate the pre-tRNA as described. The SDR5C1 base may bind two TRMT10C and RNA units (Extended Data Fig. 3d), suggesting that two pre-tRNAs or two sites within the same transcript could be processed simultaneously. PRORP is then recruited to the TRMT10C–SDR5C1–pre-tRNA subcomplex to catalyze 5′ cleavage, which may involve rearrangements to activate PRORP. Consistent with previous biochemical data[15], the structure of the mtRNase P complex does not indicate a functional coupling between RNA methylation and cleavage, and it is not clear in which order these events occur. The next steps in pre-tRNA processing are 3′ cleavage by ELAC2 and CCA addition. It has been suggested that the TRMT10C–SDR5C1 complex may serve as a platform to which the RNA remains bound for these maturation steps[29]. The structure of the TRMT10C–SDR5C1–pre-tRNA complex is consistent with this hypothesis, because it exposes both the 5′ and the 3′ cleavage sites at its distal end, where ELAC2 or other factors may bind. However, due to a lack of structural data on ELAC2, it is not clear whether this would require a factor exchange or whether ELAC2 could bind simultaneously with PRORP. Therefore, the mechanism of 3′ cleavage and how the hierarchy of reactions is ensured remains to be determined.

These results provide the framework for future studies aimed at deciphering the mechanisms of further mitochondrial RNA maturation steps and the post-transcriptional regulation of mitochondrial gene expression.

## Online content

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

## Methods

**Cloning and protein expression.** TRMT10C, SDR5C1 and PRORP genes were amplified from human complementary DNA by PCR using ligation-independent cloning-compatible primers (MRPP1 forward/reverse (for./rev.); MRPP2 for./rev.; MRPP3 for./rev.). PCR primers were designed such that the coding sequence for predicted mitochondrial targeting signals of TRMT10C (residues 1–39), SDR5C1 (1–11) and PRORP (residues 1–45) were absent from the PCR-amplified products. All PCR products were gel purified using QiaQuick Gel Extraction Kit by Qiagen. PCR products for Δ1–39 TRMT10C and Δ11 SDR5C1 were cloned into pET-derived 14-B vector (Addgene, catalog no. 48308, kind gift of S. Gradia) to generate a coexpression construct with N-terminal 6× His tag followed by a tobacco etch virus (TEV) protease cleavage site fused to Δ1–39 TRMT10C. This plasmid was further modified by adding residues 1–11 to the SDR5C1 N terminus using 'around-the-horn' PCR (MRPP2 ATH for./rev.), resulting in a coexpression vector for Δ39 TRMT10C and full-length SDR5C1.

Δ1–45 PRORP was cloned into pET-derived 14-B vector (Addgene, catalog no. 48308, kind gift of S. Gradia) with an N-terminal 6× His tag followed by a TEV protease cleavage site. All constructs were transformed into *Escherichia coli* DH5α cells for colony screening and plasmid amplification. Final expression constructs were verified by Sanger sequencing before protein expression.

Δ1-39 TRMT10C–SDR5C1 subcomplex and Δ1–45 PRORP were expressed in *E. coli* BL21 (DE3) RIL cells grown in LB media containing 100 μg ml⁻¹ ampicillin and 34 μg ml⁻¹ chloramphenicol. Cells were initially grown to an optical density of 0.5 at 600 nm at 37 °C and incubated for an additional 30 min at 16 °C. Protein expression was subsequently induced with 150 μM isopropyl-β-D-1-thiogalactopyranoside for 16 h at 16 °C. Cells were collected by centrifugation and the pellet was suspended in the lysis buffer (50 mM Na-HEPES pH 7.4 at 4 °C, 300 mM (for Δ1–39 TRMT10C–SDR5C1) or 500 mM (for Δ1–45 PRORP) NaCl, 10% glycerol, 2 mM DTT, 10–20 mM imidazole pH 8.0, 0.284 μg ml⁻¹ leupeptin, 1.37 μg ml⁻¹ pepstatin, 0.17 mg ml⁻¹ PMSF and 0.33 mg ml⁻¹ benzamidine).

**Protein purification.** All protein purification steps were carried out at 4 °C. The cells suspended in the lysis buffer were lysed by sonication and the lysate was centrifuged at 87,207*g* for 60 min. The supernatant was applied to a HisTrap HP 5 ml column (GE Healthcare) equilibrated in the lysis buffer. For Δ1–39 TRMT10C–SDR5C1 copurification, the column was washed with 9.5 column volumes (CV) of lysis buffer after sample application. For Δ1–45 PRORP purification, the column was first washed with 9.5 CV of high-salt wash buffer (50 mM Na-HEPES pH 7.5 at 4 °C, 1 M NaCl, 10% glycerol, 2 mM DTT, 20 mM imidazole pH 8.0, 0.28 μg ml⁻¹ leupeptin, 1.37 μg ml⁻¹ pepstatin, 0.17 mg ml⁻¹ PMSF and 0.33 mg ml⁻¹ benzamidine), and further washed with 9.5 CV of lysis buffer. Column-bound proteins were subsequently eluted with 9.5 CV of elution buffer (42.5 mM Na-HEPES pH 7.5 at 4 °C, 255 mM (for Δ1–39 TRMT10C–SDR5C1) or 425 mM (for Δ1–45 PRORP) NaCl, 8.5% glycerol, 1.7 mM DTT, 317 mM imidazole pH 8.0, 0.24 μg ml⁻¹ leupeptin, 1.16 μg ml⁻¹ pepstatin, 0.14 mg ml⁻¹ PMSF and 0.28 mg ml⁻¹ benzamidine). The eluate was dialyzed for 16 h against dialysis buffer (50 mM Na-HEPES pH 7.5 at 4 °C, 300 mM NaCl, 10% glycerol, 2 mM DTT) in the presence of recombinant His-tagged TEV protease (homemade: approximately 40 μg ml⁻¹ for Δ1–39 TRMT10C–SDR5C1, and approximately 60 μg ml⁻¹ for Δ1–45 PRORP). Imidazole concentration in the dialysis buffer and dialyzed eluate was adjusted to 15–20 mM. The dialyzed eluate was applied to a HisTrap HP 5-ml column equilibrated in dialysis buffer, and the column was further washed with 9.5 CV of dialysis buffer. Flow-through and wash fractions containing the protein(s) of interest cleaved by TEV protease were pooled and applied to two tandemly connected HiTrap Heparin HP 5-ml columns (effective column volume: approx. 10 ml) equilibrated with 85% (v/v) heparin affinity chromatography (HAC) buffer A (50 mM Na-HEPES pH 7.5 at 4 °C, 10% glycerol, 2 mM DTT) and 15% (v/v) HAC buffer B (50 mM Na-HEPES pH 7.5 at 4 °C, 2 M NaCl, 10% glycerol, 2 mM DTT). Column-bound proteins were eluted with a gradient from 15% to 50% v/v HAC buffer B in HAC buffer A. Fractions containing the protein of interest, identified by SDS–PAGE and Coomassie staining, were pooled and further purified using a Superdex 200 Increase 10/300 GL column equilibrated with size-exclusion buffer (20 mM Na-HEPES pH 7.5 at 4 °C, 150 mM (for Δ1–39 TRMT10C–SDR5C1) or 300 mM (Δ1–45 PRORP) NaCl, 5% (for Δ1–39 TRMT10C–SDR5C1) or 10% (for Δ1–45 PRORP) glycerol, 5 mM DTT). Peak elution fractions were analyzed using SDS–PAGE and Coomassie staining. Fractions containing the protein of interest were concentrated using Amicon Ultra-4 30K Centrifugal Filter Devices (Merck Millipore), aliquoted and flash-frozen in liquid nitrogen and stored at −80 °C until use.

**Preparation of pre-tRNA^Tyr.** The sequence of human pre-mt-tRNA^Tyr with 38 and 17 flanking nucleotides on the 5′ and 3′ end, respectively, was inserted into a modified pSP64 vector backbone[52] by Around-the-Horn PCR (tRNA^Tyr-HDV for./rev.) resulting in a construct with a 5′ T7 promoter and 3′ HDV ribozyme. The construct was transformed into *E. coli* DH5α cells for colony screening and plasmid amplification. The 3′ HDV ribozyme sequence was subsequently removed by a second around-the-horn PCR (tRNA^Tyr-RO for./rev.) resulting in a construct with 5′ T7 promoter and 3′ BamHI restriction site 7 nucleotides downstream of the insert, which yields an RNA with 38 nucleotides upstream and 24 nucleotides downstream of mt-tRNA^Tyr (38-mt-tRNA^Tyr-24) after linearization and in vitro

transcription. Plasmids were amplified in *E. coli* DH5α cells and sequence verified by Sanger Sequencing. For large-scale preparation, plasmids were transformed into *E. coli* DH5α cells and grown at 37 °C in LB media containing 100 μg ml⁻¹ ampicillin. Cells were collected from the overnight cultures by centrifugation and plasmid DNA was isolated using a NucleoBond PC 1000 Giga kit (Machery-Nagel) according to the protocol from the manufacturer. Purified plasmid DNA was linearized overnight with 3 U μl⁻¹ of BamHI-HF in 1× CutSmart Buffer (New England Biolabs). Linearized DNA was purified by phenol-chloroform extraction and isopropanol precipitation. Purified linearized plasmid templates were used for in vitro transcription (IVT) using T7 RNA polymerase (ThermoFisher). Reactions contained 1× T7 RNA Polymerase Reaction buffer, 0.001% (w/v) Triton-X 100, 30 mM MgCl₂, 4 mM NTPs, 5 U μl⁻¹ T7 RNA polymerase and 0.1 to 0.3 μg μl⁻¹ of template DNA. IVT reactions were incubated for 16 h at 37 °C and then centrifuged at 21,130*g* and 4 °C for 10 min to remove Mg-pyrophosphate precipitate. The supernatant was applied to a RESOURCE Q 6-ml column (GE Healthcare) equilibrated with anion exchange (AEX) buffer A (50 mM CH₃COO⁻Na⁺ pH 5.5, 2 mM MgCl₂) and the column was washed with 9.5 CV of AEX buffer A. Column-bound species were eluted with a 0 to 75% v/v gradient of AEX buffer B (50 mM CH₃COO⁻Na⁺ pH 5.5, 2 mM MgCl₂, 2 M NaCl) in AEX buffer A. Peak eluate fractions were analyzed by denaturing gel electrophoresis (urea-PAGE) using 8 M urea, 7% polyacrylamide (bis-acrylamide to acrylamide ratio 1:29) gels in 1× Tris-borate-EDTA buffer, and subsequent staining with methylene blue. Fractions containing the product RNA were pooled and applied to PD-10 desalting columns (GE Healthcare) equilibrated with 300 mM CH₃COO⁻Na⁺ pH 5.5 for buffer exchange. The RNA was precipitated in isopropanol, dissolved in nuclease-free water and stored at −20 °C until use.

**RNase P cleavage assays.** All RNase P cleavage assays were carried out in 20-μl reactions containing RNA cleavage assay buffer (100 mM Tris-HCl pH 8.0 at 4 °C, 100 mM NaCl, 4 mM DTT) and 200 nM pre-tRNA^Tyr (38-mt-tRNA^Tyr-24). Wherever present, Δ1–39 TRMT10C–SDR5C1 complex and Δ1–45 PRORP were present at 1 μM final concentration. MgCl₂ and NADH were present at final concentrations of 5 mM and 50 μM, respectively, unless otherwise specified. For RNase P cleavage assays, the RNA cleavage assay buffer, MgCl₂, NADH and Δ1–39 TRMT10C–SDR5C1 complex were first mixed and incubated at 30 °C for 5 min. Pre-tRNA was added and reactions were incubated at 30 °C for a further 15 min. Δ1–45 PRORP was added, and reactions were incubated at 30 °C for a further 60 min. The reactions were quenched with an equal volume of 2× RNA loading buffer (6.4 M urea, 2× Tris-borate-EDTA buffer, 50 mM EDTA pH 8.0) and 1.6 U of recombinant proteinase K (New England Biolabs) was added. The reactions were incubated at 30 °C for 60 min and stored at −20 °C until analysis by urea–PAGE. Before urea–PAGE, the reactions were thawed and incubated at 95 °C for 3 min. A 10-μl portion of each reaction was loaded onto 8 M urea, 7% polyacrylamide (bis-acrylamide to acrylamide ratio 1:29) denaturing gels in 1× Tris-borate-EDTA buffer. After electrophoresis, gels were stained with SYBR Gold nucleic acid stain and imaged using a Typhoon FLA 9500 laser scanner (GE Healthcare). All assays were performed in three independent replicates. Uncropped gels are shown in source data for Extended Data Fig. 1.

**Cryo-EM sample preparation and data collection.** The full mtRNase P complex was assembled in 1× complex buffer (100 mM Tris-HCl pH 8.0 at 4 °C, 100 mM NaCl, 250 μM EDTA, 2 mM DTT) in a 200-μl total reaction volume. For complex formation, 3.36 nmol of Δ1–39 TRMT10C–SDR5C1 complex was mixed with a 2-fold molar excess of the pre-tRNA^Tyr and a 30-fold molar excess of NADH and incubated at 30 °C for 15 min. A 2.5-fold molar excess of PRORP was added and the reaction mix was incubated at 30 °C for a further 30 min. The reaction mixture was applied to a Superdex 200 Increase 3.2/300 column (GE Healthcare) equilibrated in complex purification buffer (40 mM Na-HEPES pH 7.5 at 4 °C, 150 mM NaCl, 0.25 mM EDTA, 2 mM DTT). Eluate fractions were analyzed by SDS–PAGE and Coomassie staining. Peak fractions were additionally analyzed by denaturing gel electrophoresis in 8 M urea, 7% polyacrylamide gels in 1× TBE buffer and methylene blue staining. The peak fraction was diluted with complex purification buffer to an absorbance at 280 nm of 1.5 AU (path length 10 mm). 4-μl of diluted complex were applied to freshly glow-discharged R 2/1 holey carbon grids (Quantifoil), and the grids were blotted with a blot force of 5 for 4 s using a Vitrobot Mark IV (ThermoFisher) at 4 °C and 100% humidity immediately before plunge-freezing in liquid ethane.

Cryo-EM data were collected on a 300 keV Titan Krios cryo-transmission electron microscope (FEI Company) equipped with a K3 electron counting direct detector (Gatan) and a GIF quantum energy filter, operated using Serial EM[53]. Images were acquired at 105,000-fold magnification, corresponding to a calibrated pixel size of 0.834 Å, in EFTEM mode with an energy filter slit width of 20 eV and saved as non-super-resolution counting image stacks of 40 movie frames, with an electron dose of 0.99 e⁻ Å⁻² per frame. Warp was used for on-the-fly preprocessing of the collected movies, including gain-reference correction, dose weighting, motion correction, frame alignment and CTF estimation[54]. The generic particle picker neural network in Warp, BoxNet, was retrained with a dataset of manually picked particles from ten micrographs, and used for particle picking. Picked particles were extracted using Warp with a pixel size of 2.835 Å.

**Cryo-EM data processing and analysis.** A total of 6.15 million particles picked and extracted by Warp were exported to cryoSPARC[55] for 2D classification. A total of 1.86 million particles from the manually selected 'good' classes were used for 3D classification using the heterogeneous refinement algorithm in cryoSPARC. A model of the TRMT10C–SDR5C1–pre-tRNA[Tyr] complex generated from a dataset previously collected on a 200 keV Glacios Cryo-TEM microscope and processed using cryoSPARC was used as the initial reference model for the 3D classification. A total of 498,269 particles from the class resembling the TRMT10C–SDR5C1–pre-tRNA complex with additional density were further refined in 3D, and the refined model, as well as the initial 1.86 million particles selected from the 2D classification, were exported to RELION v.3.0 (ref. [56]). Exported particles were subjected to 3D classification in RELION v.3.0 using the model obtained from cryoSPARC as the initial reference, and 493,603 particles, corresponding to the 'good' class, were subjected to a second of 3D classification using the model for the corresponding class in the earlier classification as the initial reference. A total of 113,575 particles were selected from the class containing an additional density, along with the density for TRMT10C–SDR5C1–pre-tRNA complex, and re-extracted in RELION without binning and a 400-pixel box size. Re-extracted particles were subjected to 3D classification with local alignment with a 3D mask around the additional density. Particles selected from the class with strong density (88,081 particles) within the masked volume were subjected to CTF refinement and Bayesian polishing in RELION. Polished particles were subjected to a global 3D refinement and a focused 3D refinement using a 3D mask around the additional density. After post-processing in RELION, the global (map 1) and focused refinement (map 2) maps were reported to be at average resolutions of 3.0 Å and 3.1 Å, respectively.

To investigate the poorly resolved density on the opposite face of SDR5C1 base, focused 3D classification using a mask encompassing the poor density was carried out. The resulting, approximately equally distributed particle populations, were subjected to independent 3D refinements, which resulted in map 3 and map 4 (Extended Data Fig. 3).

**Model building and refinement.** To obtain an initial model, previous crystal structures of the SDR5C1 tetramer (PDB 1U7T)[22], Δ1–200 TRMT10C (methyltransferase domain) (PDB 5NFJ)[19], Δ206 PRORP (PDB 4XGL)[23] and yeast mitochondrial tRNA[Phe] (PDB 3TUP)[57] were rigid-body-fit in the final maps using UCSF Chimera[58]. Fitted models were subsequently rebuilt and refined in real space in Coot[59]. TRMT10C, SDR5C1 and the majority of the tRNA[Tyr] were modeled using the global refinement map (map 1), while PRORP and parts of the pre-tRNA[Tyr] were modeled using the focused refinement map (map 2). Residues 92–156 and 175–202 of TRMT10C and residues 117–207 of PRORP, and the N-terminal domain of TRMT10C, were modeled de novo. The sequence of yeast mt-tRNA[Phe] was adjusted to that of human mt-tRNA[Tyr] by iteratively mutating, deleting or adding nucleotides along the tRNA sequence. A composite map was generated from the global refinement map (map 1) and the focused refinement map (map 2) using phenix.combined_focused_maps[60]. The complete model of the mtRNase P complex was first real space refined against the composite map using phenix.real_space_refine[60,61], followed by a single round of ADP refinement against the global refinement map. Validation was carried out with the composite map using the MolProbity[62] package within the Phenix suite. This final model showed excellent stereochemistry (Table 1). Figures were prepared with ChimeraX[63] and PyMOL (The PyMOL Molecular Graphics System, v.1.2r3pre, Schrödinger, LLC). Potential hydrogen bonds, shown as black dashed lines, were predicted using the 'hbonds' command in ChimeraX[64] with tolerance of 0.5 Å and 45° for distance and angles, respectively. Surface potential renderings were generated in ChimeraX.

To analyze binding of a second copy of TRMT10C and substrate to the SDR5C1 tetramer, the two reconstructions obtained after focused classification and refinement (map 3 and map 4) were used to rigid-body-fit the CTD and adapter domain of TRMT10C (residues 175–385). This revealed that the two reconstructions correspond to two binding orientations of TRMT10C (Extended Data Fig. 3). In addition, diffuse density that may correspond to a second pre-tRNA substrate was observed. Due to the limited quality of these maps, we refrained from any further modeling.

**Reporting Summary.** Further information on research design is available in the Nature Research Reporting Summary linked to this article.

## Data availability
The electron potential reconstructions and structure coordinates were deposited with the Electron Microscopy Database (EMDB) under accession code EMD-13002 and with the Protein Data Bank (PDB) under accession code PDB 7ONU. Source data are provided with this paper.

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

## Acknowledgements
H.S.H. was supported by the Deutsche Forschungsgemeinschaft (FOR2848, SFB1190, EXC 2067/1-390729940). P.C. was supported by the Deutsche Forschungsgemeinschaft (SFB860, SPP2191, EXC 2067/1-390729940) and the ERC Advanced Investigator Grant CHROMATRANS (grant agreement No 693023). Funded by the Deutsche Forschungsgemeinschaft (DFG, German Research Foundation) under Germany's Excellence Strategy EXC 2067/1- 390729940.

## Author contributions
A.B. carried out all experiments. C.D. assisted with cryo-EM data acquisition. P.C. provided funding and conceptual input. H.S.H. conceived and supervised the project and acquired funding. A.B., P.C. and H.S.H. wrote the manuscript with input from C.D.

## Funding

## Competing interests
The authors declare no competing interests.

## Additional information
**Extended data** is available for this paper at https://doi.org/10.1038/s41594-021-00637-y.

**Correspondence and requests for materials** should be addressed to Hauke S. Hillen.

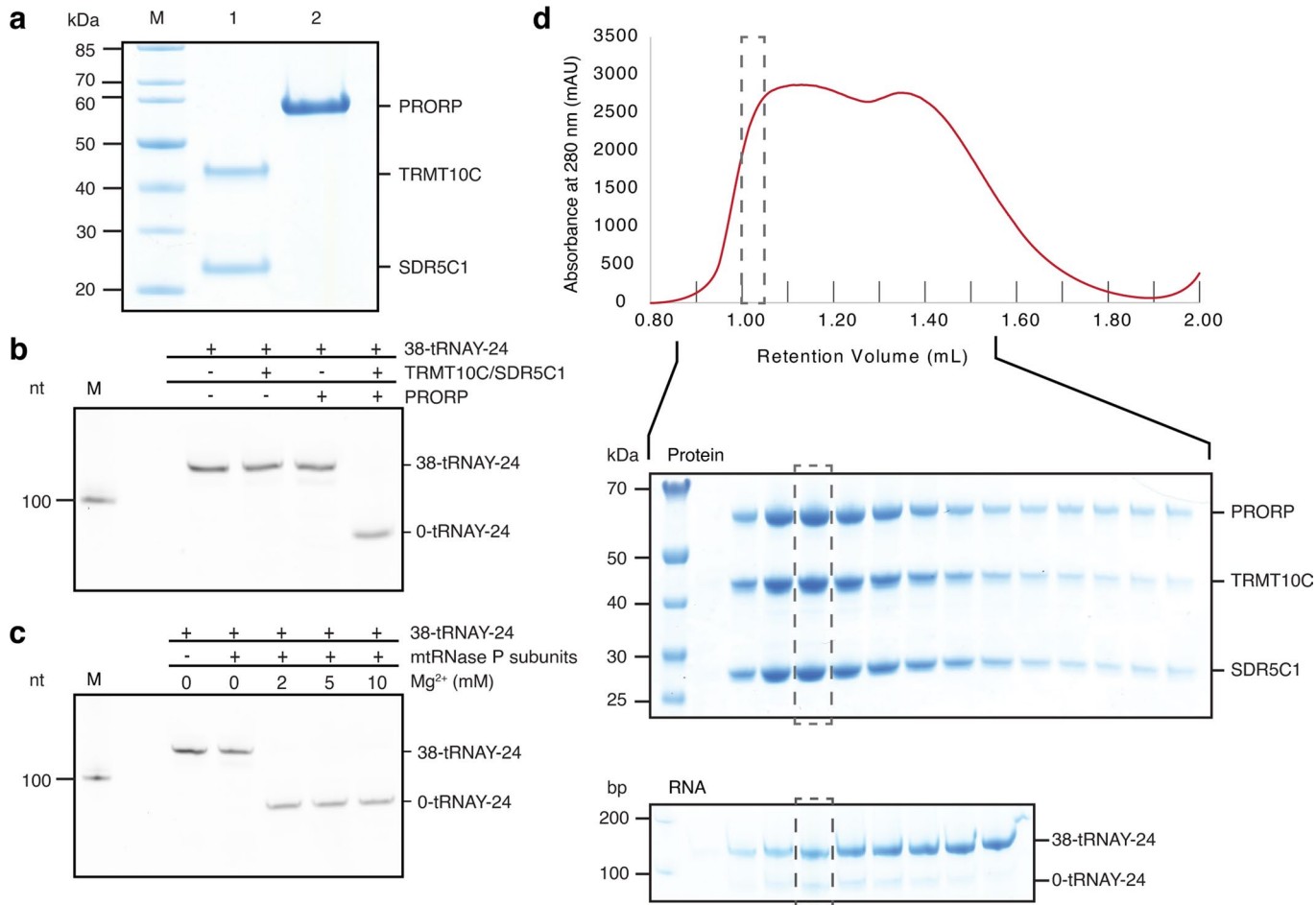

**Extended Data Fig. 1 | Reconstitution of human mtRNase P. a**, SDS PAGE analysis of purified recombinant mtRNase P complex subunits. Lanes: M – protein molecular weight marker, 1 – co-purified Δ39-TRMT10C-SDR5C1 complex, 2 – Δ45-PRORP. Approximately 300 ng of purified proteins were loaded in lanes 1 and 2. **b**, Cleavage of pre-tRNA^Tyr by mtRNase P subunits and full mtRNase P complex. Lanes from left to right: RNA size marker, empty lane, only pre-tRNA, pre-tRNA with TRMT10C-SDR5C1, pre-tRNA with PRORP, and pre-tRNA with TRMT10C-SDR5C1 + PRORP. **c**, Effect of Mg²⁺ on pre-tRNA^Tyr cleavage by the mtRNase P complex. First three lanes from left: RNA size marker, empty lane, pre-tRNA without mtRNase P subunits or Mg²⁺. Subsequent lanes contain pre-tRNA + TRMT10C-SDR5C1 + PRORP and varying concentrations of Mg²⁺ as indicated. **d**, *In-vitro* reconstitution and purification of the mtRNase P complex by size exclusion chromatography (SEC) in Mg²⁺-depleted conditions. The SEC chromatogram between 0.80 and 2.00 mL retention volumes is shown. Eluate fractions between 0.85 and 1.55 mL were analyzed by SDS PAGE to assess protein content; SEC eluate fractions between 0.85 and 1.30 mL were additionally analyzed by Urea PAGE to assess RNA content. Fractions within the gray dashed rectangle (1.00–1.05 mL) were used for single particle analysis by cryo-EM. Additional replicates for panels b and c, along with uncropped images of all gels, and raw data for the chromatogram in panel **d** are available as source data.

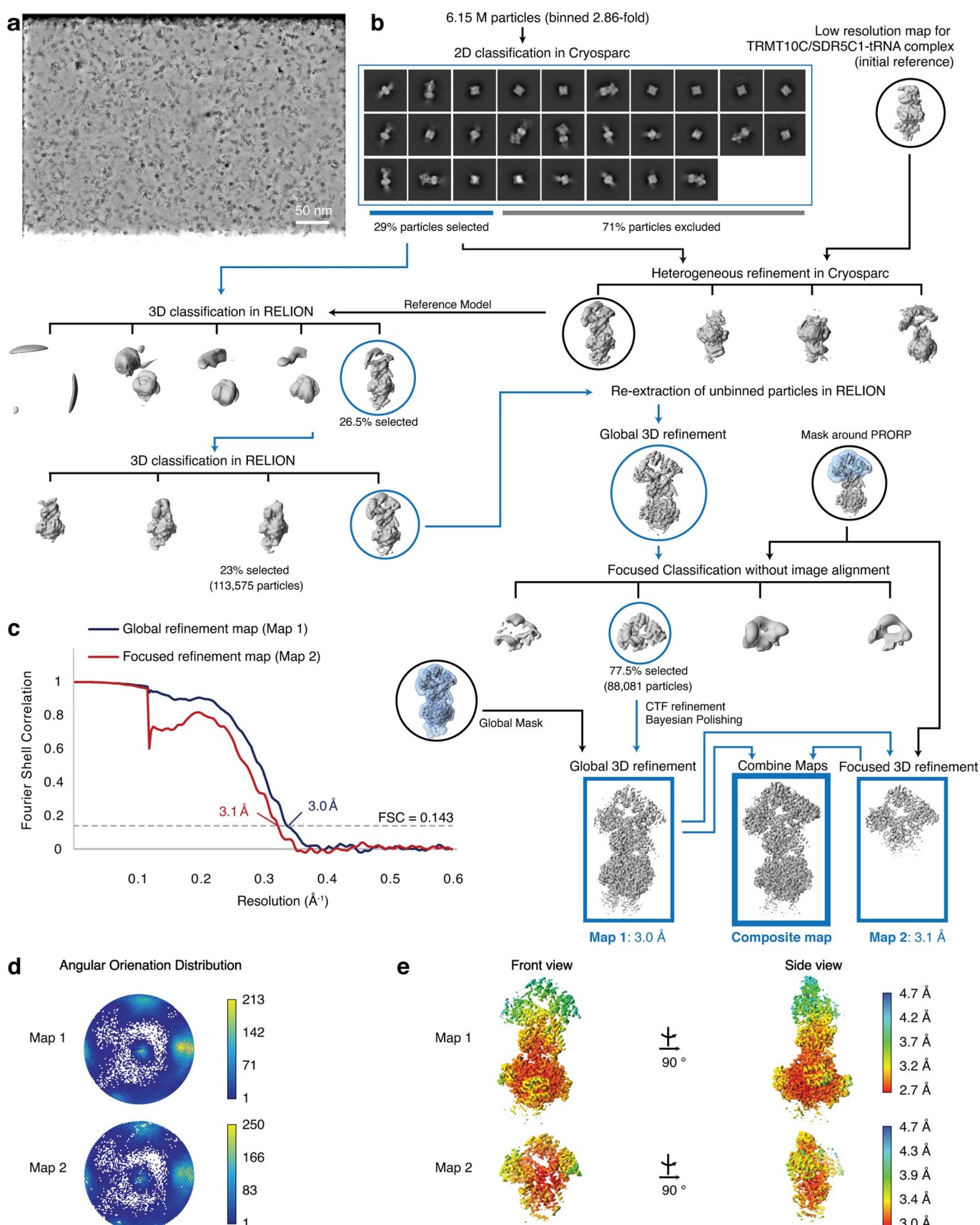

**Extended Data Fig. 2 | Cryo-EM processing. a**, Example denoised micrograph calculated from two independently measured half sets of 40 frames each. Scale bar, 50 nm. **b**, 2D classes & processing tree. **c**, Fourier Shell Correlation (FSC) plots for both maps. **d**, Angular Distribution Plots calculated with Warp. **e**, Local Resolution Maps calculated in RELION.

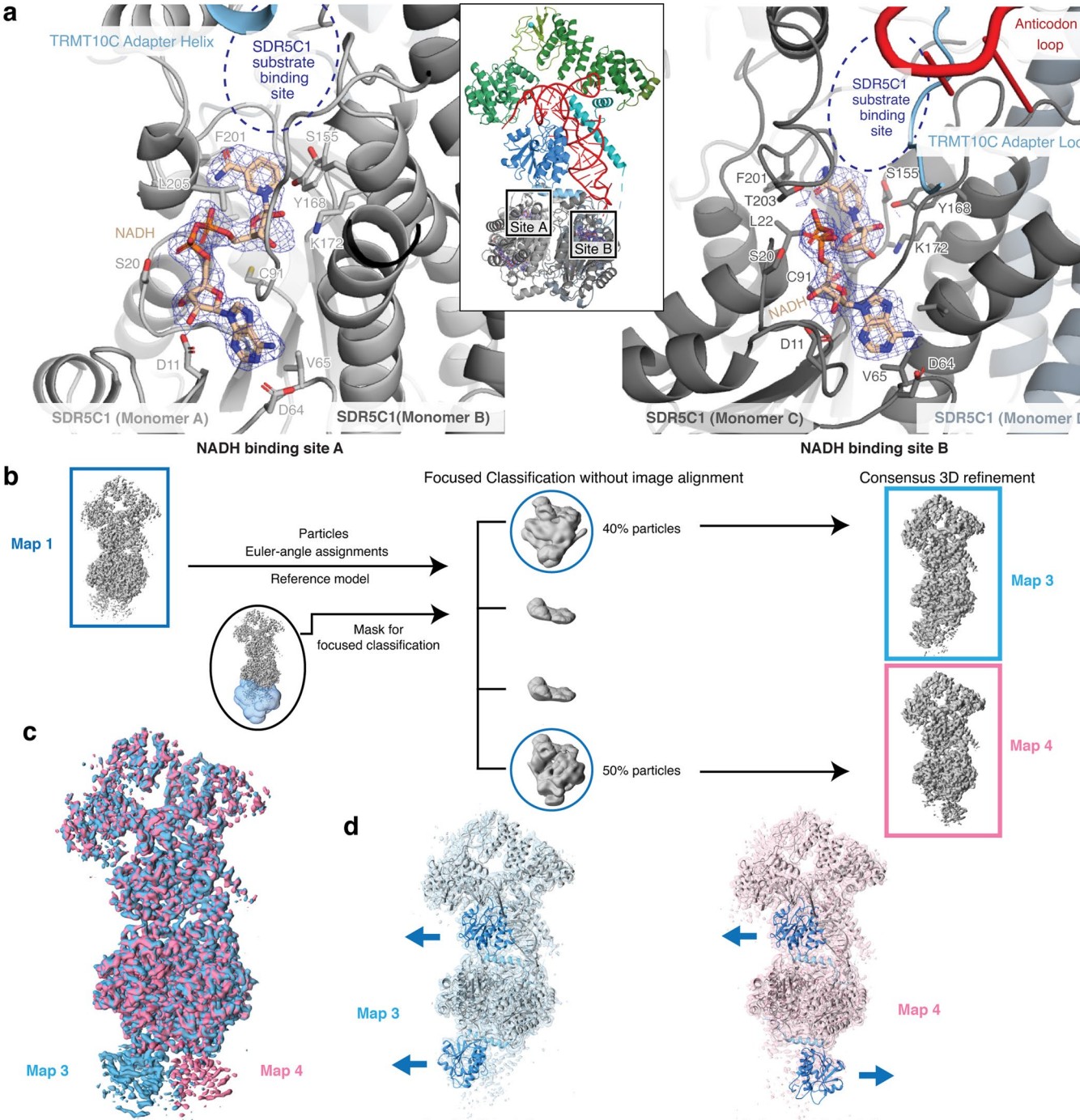

**Extended Data Fig. 3 | Structural details of the TRMT10C-SDR5C1 subcomplex. a**, NADH bound to the two non-equivalent NADH binding sites in the SDR5C1 tetramer. NADH and residues within 3.5 Å are shown as sticks. The cryo-EM density around NADH is shown as blue mesh. The NADH binding site A (left) is located near the C-terminal end of the TRMT10C adapter helix, while the NADH binding side B (right) is located near the TRMT10C adapter loop and anticodon recognition site. The dehydrogenase substrate binding sites near both NADH binding sites are partially occluded by the adapter region of TRMT10C and/or tRNA. **b**, Cryo-EM processing workflow to resolve the two opposing orientations of the second TRMT10C monomer bound to SDR5C1 tetramer. **c**, Overlay of the two reconstructions map 3 and map 4, showing opposing orientations of the bottom density. Surface color of the maps correspond to the color used for boxes around map 3 and map 4 in (b). **d**, Models of mtRNase P with a second TRMT10C monomer (residues 175–385) bound to SDR5C1 tetramer in two opposing orientations fitted into cryo-EM density maps from (b) colored as in (c). Domain coloring of TRMT10C between residues 175 to 385 as in Fig. 1(d); models for all remaining regions are shown in gray. The adapter helix of TRMT10C can bind to the groove on SDR5C1 tetramer in two 180°-rotated orientations resulting in either parallel orientation or antiparallel orientation of TRMT10C protomers. Arrows next to each TRMT10C monomer denote the general orientation of the N and C termini, with the arrowhead denoting the C-terminus.

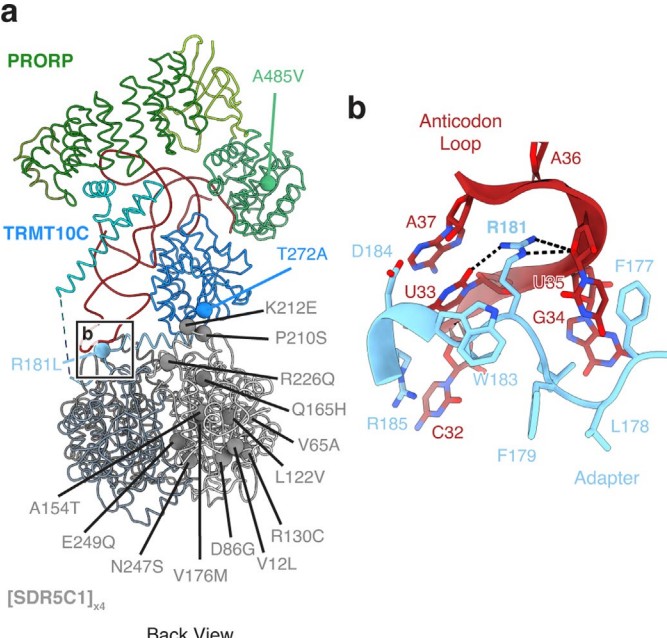

**Extended Data Fig. 4 | Structural mapping of disease mutations. a**, Structural mapping of reported disease mutations in mtRNase P complex subunits[34,65-67]. The structure of mtRNase P is shown in wire representation. Cβ atoms of mutated residues are shown as spheres. The mutation site shown enlarged in panel (b) is indicated. **b**, Close-up view of TRMT10C R181L mutation site. RNA nucleotides and residues within 4 Å of R181 are shown as sticks. R181 protrudes into the distorted anticodon loop and interacts with U33. Mutation of this residue likely leads to impaired tRNA recognition.

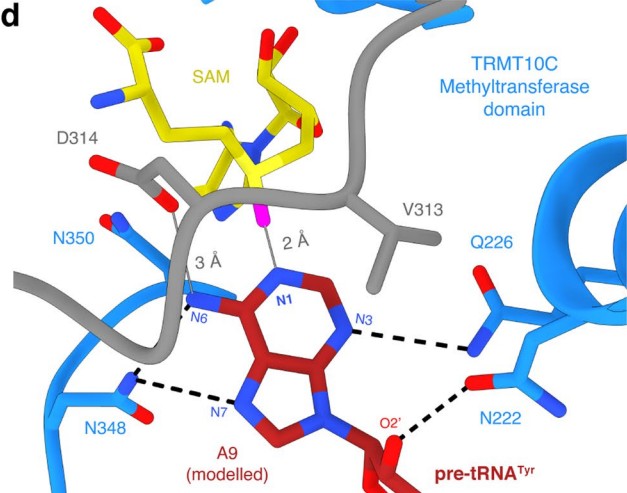

**Extended Data Fig. 5 | Comparison between apo and active TRMT10C methyltransferase domains. a**, Electrostatic surface potential for TRMT10C methyltransferase and adapter domains. Electrostatic surface coloring as in Fig. 5(c). The tRNA backbone, shown in red, interacts with positively charged regions on the TRMT10C surface. **b**, Close-up view of the methyltransferase active site of TRMT10C without SAM in the mtRNase P complex. Depiction as in Fig. 4(b). Motif II of TRMT10C (residues 314–319, shown in gray) adopts an open conformation. Arrows approximate the rearrangement of motif II residues upon SAM binding. **c**, Close-up of the methyltransferase active site of TRMT10C in the SAM-bound crystal structure (PDB: 5NFJ)[19]. Motif II of TRMT10C adopts a closed conformation. **d**, Pre-catalytic model of TRMT10C methyltransferase active site for A9 methylation. Adenine was modeled in the active site by changing the atomic coordinates of guanine to adenine while preserving the base conformation. TRMT10C and SAM are represented as in Fig. 4(c). Hydrogen bonds are shown as dashed black lines. Distances between A9–N1 and the SAM–methyl group, and between A9–N6 and the D314–sidechain carbonyl are labeled and shown as solid graey lines.

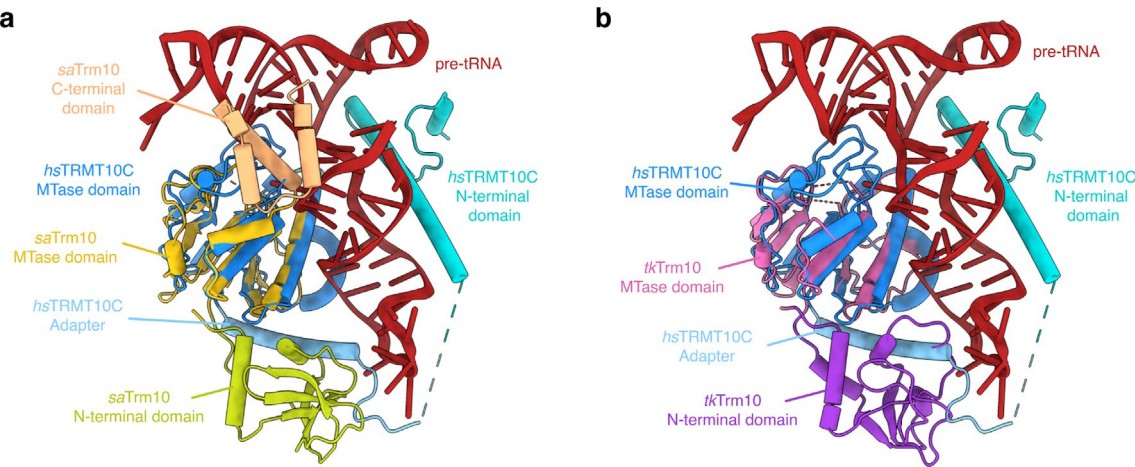

**Extended Data Fig. 6 | Structural comparison of TRMT10C homologs. a-b**, Structural comparison of human TRMT10C (*hs*TRMT10C) in the mtRNase P complex and (a) full-length *Sulfolobus acidocaldarius* Trm10 (*sa*Trm10) (PDB code: 5A7Y)[35] and (b) full-length *Thermococcus kodakaraensis* Trm10 (*tk*Trm10) (PDB code: 6EMU)[36]. The structures are superimposed on the respective methyltransferase domains. Human TRMT10C domains are colored as in Fig. 1(a). For *sa*Trm10 (a), the N-terminal domain is colored in lime, the methyltransferase domain in yellow, and the C-terminal domain in light orange. For *tk*Trm10 (b), the N-terminal domain is colored in purple and the MTase domain is colored in pink.

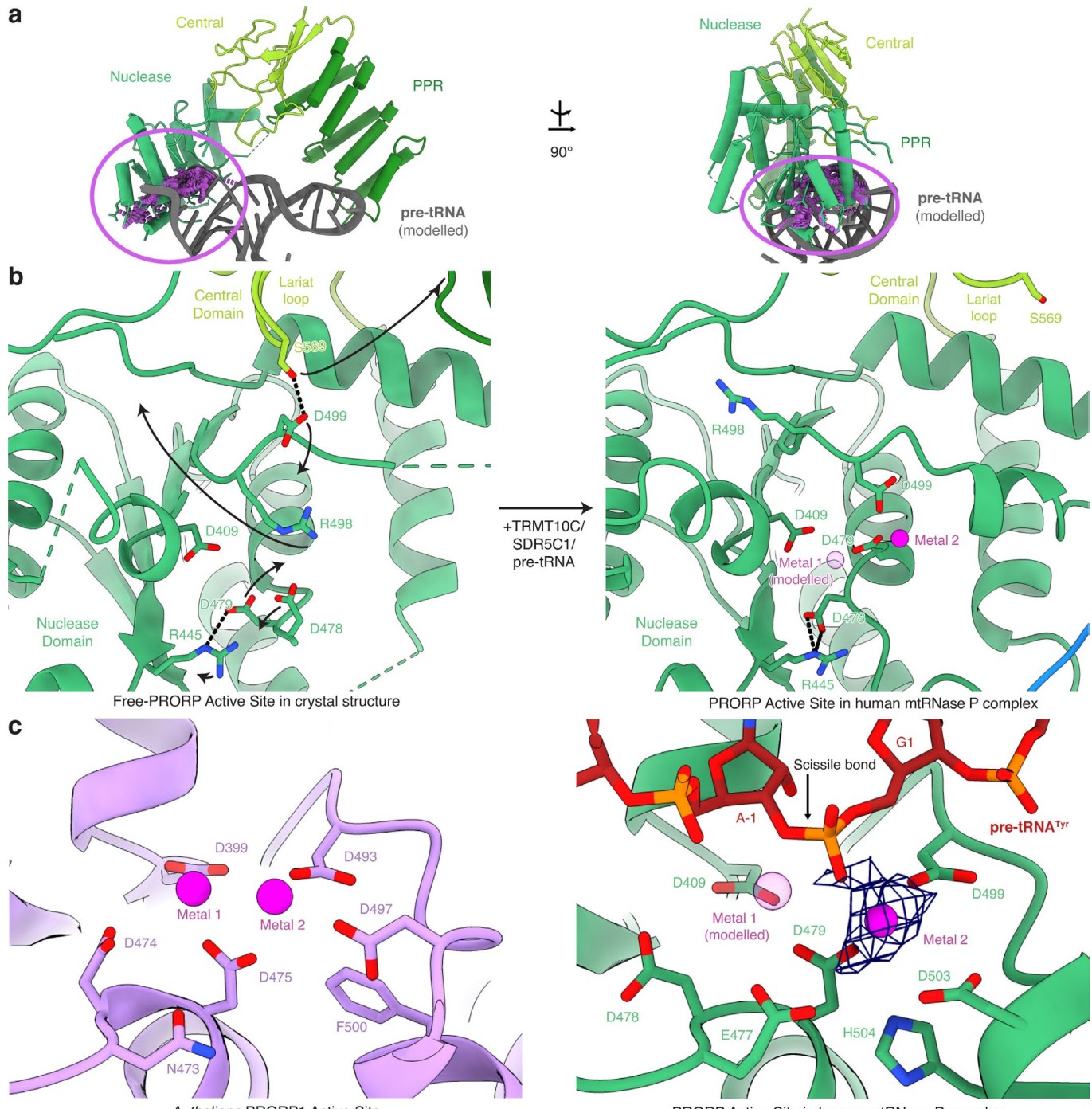

**Extended Data Fig. 7 | Comparison of inactive and active conformations of PRORP. a,** Clashes between metallonuclease domain of free PRORP in the closed conformation and the substrate pre-tRNA. The crystal structure of free PRORP (PDB: 4XGL)[23] was superimposed via its PPR domain to PRORP in mtRNase P complex. Pre-tRNA^Tyr, shown in gray, was modeled based on its binding site in mtRNase P complex. Clashes between pre-tRNA and free PRORP are indicated in purple. **b,** Comparison of the PRORP nuclease active site observed in the crystal structure of free PRORP (PDB: 4XGL)[23] and in the active conformation in the mtRNase P complex. Arrows indicate the rearrangements that may occur in the free PRORP structure upon binding to the TRMT10C-SDR5C1 complex and substrate RNA. Active site aspartates are shown as sticks. Metal 1 was modeled based on Mn²⁺ binding sites in the At-PRORP1 crystal structure (PDB: 4G24)[41]. The pre-tRNA is omitted for clarity. **c,** Comparison of the active sites of At-PRORP1 (PDB: 4G24)[41] (left) and active PRORP (right). The two metal ions in the At-PRORP1 structure are shown as spheres. The cryo-EM density map for metal 2 in the PRORP active site is shown as blue mesh.

# Reporting Summary

Nature Research wishes to improve the reproducibility of the work that we publish. This form provides structure for consistency and transparency in reporting. For further information on Nature Research policies, see our Editorial Policies and the Editorial Policy Checklist.

## Statistics

For all statistical analyses, confirm that the following items are present in the figure legend, table legend, main text, or Methods section.

| n/a | Confirmed | |
|---|---|---|
| ☒ | ☐ | The exact sample size (*n*) for each experimental group/condition, given as a discrete number and unit of measurement |
| ☒ | ☐ | A statement on whether measurements were taken from distinct samples or whether the same sample was measured repeatedly |
| ☒ | ☐ | The statistical test(s) used AND whether they are one- or two-sided *Only common tests should be described solely by name; describe more complex techniques in the Methods section.* |
| ☒ | ☐ | A description of all covariates tested |
| ☒ | ☐ | A description of any assumptions or corrections, such as tests of normality and adjustment for multiple comparisons |
| ☒ | ☐ | A full description of the statistical parameters including central tendency (e.g. means) or other basic estimates (e.g. regression coefficient) AND variation (e.g. standard deviation) or associated estimates of uncertainty (e.g. confidence intervals) |
| ☒ | ☐ | For null hypothesis testing, the test statistic (e.g. *F*, *t*, *r*) with confidence intervals, effect sizes, degrees of freedom and *P* value noted *Give P values as exact values whenever suitable.* |
| ☒ | ☐ | For Bayesian analysis, information on the choice of priors and Markov chain Monte Carlo settings |
| ☒ | ☐ | For hierarchical and complex designs, identification of the appropriate level for tests and full reporting of outcomes |
| ☒ | ☐ | Estimates of effect sizes (e.g. Cohen's *d*, Pearson's *r*), indicating how they were calculated |

*Our web collection on statistics for biologists contains articles on many of the points above.*

## Software and code

Policy information about availability of computer code

| Data collection | Serial EM 3.8 beta 8 |
|---|---|
| Data analysis | RELION 3.0 beta-2, UCSF Chimera 1.13, UCSF ChimeraX v0.8, Pymol 1.2r3pre, Coot 0.9, Warp v1.0.7, PHENIX 1.19.2-4158, cryoSPARC 2.14.2 |

For manuscripts utilizing custom algorithms or software that are central to the research but not yet described in published literature, software must be made available to editors and reviewers. We strongly encourage code deposition in a community repository (e.g. GitHub). See the Nature Research guidelines for submitting code & software for further information.

## Data

Policy information about availability of data

All manuscripts must include a data availability statement. This statement should provide the following information, where applicable:

- Accession codes, unique identifiers, or web links for publicly available datasets
- A list of figures that have associated raw data
- A description of any restrictions on data availability

The electron potential reconstructions and structure coordinates were deposited with the Electron Microscopy Database (EMDB) under accession code EMD-13002 and with the Protein Data Bank (PDB) under accession code PDB 7ONU. Source data for Extended Data Figure 1 are provided with this paper.

# Field-specific reporting

Please select the one below that is the best fit for your research. If you are not sure, read the appropriate sections before making your selection.

☒ Life sciences          ☐ Behavioural & social sciences          ☐ Ecological, evolutionary & environmental sciences

For a reference copy of the document with all sections, see nature.com/documents/nr-reporting-summary-flat.pdf

# Life sciences study design

All studies must disclose on these points even when the disclosure is negative.

| | |
|---|---|
| Sample size | No statistical methods were used to predetermine sample size, as the results of biochemical experiments were not statistically analyzed. The sample size (number of particles) for single-particle cryo-electron microscopy was determined by the number of particle images that could be acquired in a defined time slot. |
| Data exclusions | No data were excluded from the analyses. |
| Replication | All attempts at replication were successful. Biochemical cleavage assays were repeated as technical triplicates, and the results for each replicate are provided as Source Data. |
| Randomization | Samples were not allocated to groups. |
| Blinding | Investigators were not blinded during data acquisition and analysis because it is not a common procedure for the methods employed and does not effect the outcome of the experiments. |

# Reporting for specific materials, systems and methods

We require information from authors about some types of materials, experimental systems and methods used in many studies. Here, indicate whether each material, system or method listed is relevant to your study. If you are not sure if a list item applies to your research, read the appropriate section before selecting a response.

### Materials & experimental systems

| n/a | Involved in the study |
|---|---|
| ☒ ☐ | Antibodies |
| ☒ ☐ | Eukaryotic cell lines |
| ☒ ☐ | Palaeontology and archaeology |
| ☒ ☐ | Animals and other organisms |
| ☒ ☐ | Human research participants |
| ☒ ☐ | Clinical data |
| ☒ ☐ | Dual use research of concern |

### Methods

| n/a | Involved in the study |
|---|---|
| ☒ ☐ | ChIP-seq |
| ☒ ☐ | Flow cytometry |
| ☒ ☐ | MRI-based neuroimaging |

