## [Peer Review File · Nature Structural & Molecular Biology]

Peer Review Information

Journal: Nature Structural and Molecular Biology

Manuscript Title: Structural basis of RNA processing by human mitochondrial RNase P

Corresponding author name(s): Hauke Hillen

Reviewer Comments & Decisions:

Decision Letter, initial version:
--

13th Apr 2021

Dear Hauke,

Thank you again for submitting your manuscript "Structural basis of RNA processing by human mitochondrial RNase P". We have now received the comments from three reviewers who assessed your study (comments appended below). In light of those reports, we remain interested in your study and would like to invite you to respond to the comments of the referees, in the form of a revised manuscript.

You will see that all reviewers appreciate the structure of human mitochondrial RNase P in complex with a mitochondrial tRNA precursor. Nevertheless, both reviewer #1 and reviewer #3 raise a number of questions regarding the data interpretation and request textual and presentational revisions. Please be sure to address/respond to all concerns of the referees in full in a point-by-point response and highlight all changes in the revised manuscript text file. If you have comments that are intended for editors only, please include those in a separate cover letter.

(Please note in reference to the final comment of reviewer #1 that the NSMB article format does require a discussion section.)

We expect to see your revised manuscript within 6 weeks. If you cannot send it within this time, please contact us to discuss an extension; we would still consider your revision, provided that no similar work has been accepted for publication at NSMB or published elsewhere.

Reporting Summary:

Please note that all key data shown in the main figures as cropped gels or blots should be presented in uncropped form, with molecular weight markers. These data can be aggregated into a single supplementary figure item. While these data can be displayed in a relatively informal style, they must refer back to the relevant figures. These data should be submitted with the final revision, as source data, prior to acceptance, but you may want to start putting it together at this point.

Data availability: this journal strongly supports public availability of data. All data used in accepted papers should be available via a public data repository, or alternatively, as Supplementary Information. If data can only be shared on request, please explain why in your Data Availability Statement, and also in the correspondence with your editor. Please note that for some data types, deposition in a public repository is mandatory - more information on our data deposition policies and available repositories can be found below:

<https://www.nature.com/nature-research/editorial-policies/reporting-standards#availability-of-data>

We require deposition of coordinates (and, in the case of crystal structures, structure factors) into the Protein Data Bank with the designation of immediate release upon publication (HPUB). Electron

microscopy-derived density maps and coordinate data must be deposited in EMDDB and released upon publication. To avoid delays in publication, dataset accession numbers must be supplied with the final accepted manuscript and appropriate release dates must be indicated at the galley proof stage.

[REDACTED]

With kind regards,
Anke

Anke Sparmann, PhD
Senior Editor
Nature Structural and Molecular Biology
ORCID 0000-0001-7695-2049

Referee expertise:

Referee #1: mitochondrial tRNA processing

Referee #3: RNA-protein structure

Reviewers' Comments:

Reviewer #1:

Remarks to the Author:

The manuscript by Bhatta et al. reports the cryo-EM structure of human mitochondrial RNase P in complex with a mitochondrial tRNA precursor. Although the study does not include any accompanying biochemical analyses to substantiate its mechanistic implications/conclusions, the mere structure on its own represents a major scientific achievement in the field. It represents the first structure of a proteinaceous RNase P in complex with its pre-tRNA substrate and at the same time also the first

structure of a TRM10-type methyltransferase with its tRNA substrate. Together with previously published crystallographic studies of individual proteins or protein domains, as well as biochemical studies, the cryo-EM structure of the holoenzyme-substrate complex allows a new level of insight and will certainly inspire the design of future studies testing its mechanistic and evolutionary implications. While this new structure thus in my eyes deserves being published, the current report falls short in appropriately putting the data in the context of previously published work and makes mechanistic and evolutionary claims that are not (sufficiently) supported by the available structural information.

References in the following are either indicated by the reference number used in the manuscript or, if new, by their DOI.

The authors conclude that (lines 188-9)“the structure of the mtRNase P complex provides the mechanistic basis for N1-methylation at position 9 of tRNAs.” However, although the structure nicely shows the expected rotation of the purine of position 9 out of the tRNA core and some of its interactions within the active site, the authors don’t even attempt to address the actual mechanism of methylation. Notably, the mechanism of methylation by the TRM10 class of methyltransferases has been elusive and remains unclear despite active-site mutagenesis studies and crystal structures of several TRM10 methyltransferases, all of them without a tRNA substrate, however. Given the latter limitation of previous studies, the presented structure could indeed be expected to provide some insight into the catalytic mechanism, despite the lack of the SAM cofactor, yet the issue is not actually touched on in the manuscript.

Moreover, a comparison of the MRPP1 (TRMT10C) structure to the 2 previously published full-length TRM10 structures (10.1093/nar/gkv1369; 10.1261/rna.064345.117) could be informative. Other than suggested by the misleading introduction of TRMT10C’s gross structure (lines 48-51), all TRM10 enzymes appear to have a homologous N-terminal domain (NTD) in addition to their SPOUT methyltransferase domain. This NTD appears thus not specifically required for the pre-tRNA-processing role of TRMT10C in the mtRNase P complex, but apparently for the interaction of TRMT10C with the tRNA (as also demonstrated by the presented structure). A comparison of the available structures could indicate whether other TRM10 enzymes possibly interact in a similar way with their tRNA substrates. How do their NTD structures compare? Do they all contain a “connector” helix lined by basic residues, like TRMT10C? Are the connections between the SPOUT domains and the NTDs of other TRM10 enzymes flexible enough to encase a tRNA in between them in a similar way? Such a comparison could finally also provide some insight in the unique dependence of TRMT10C on the MRPP2 tetramer for methylation, a key question that is not sufficiently addressed by the current analysis of the mtRNase P structure.

The discussion of the recruitment and activation of the nuclease subunit, a supposedly dynamic process, is based on a comparison to 2 previously reported crystal structures of MRPP3 fragments and entirely follows the scenario suggested in those papers (refs. 14 and 15). However, others have more recently raised substantial criticism in this interpretation (10.3390/biom6020027; 10.1007/978-3-030-57246-4_11). As the new structure does not resolve the concerns raised and no additional biochemical or structural data are presented that would address this point, I think the discussion should more critically address the issue of the “activation” of the nuclease subunit. E.g., sentences like “In the previous crystal structures of apo-MRPP3, the enzyme adopted an auto-inhibited conformation ...” (line 195) appear misleading, as the previously crystallized fragments were enzymatically inactive even in the presence of the MRPP1/MRPP2 (thus neither “apo-MRPP3” nor “enzyme” appear appropriate in this context). In fact, the cryo-EM structure now demonstrates that those MRPP3

fragments were obviously inactive due to the deletion of the N-terminal part of the PPR domain, which is shown to be involved in the interaction of MRPP3 with the pre-tRNA and MRPP1; similar N-terminal deletions of single-subunit PRORPs were likewise inactive, by the way. "This explains why this region of MRPP3 is critical for activity although it was not resolved in the previous MRPP3 crystal structure" (lines 205-7) is thus also misleading, because the addressed region was simply not included in the crystalized MRPP3 fragments. As discussed previously (10.3390/biom6020027; 10.1007/978-3-030-57246-4_11) it still appears equally reasonable that the distortion of the active site is related to the N-terminal deletion, rather than representing a physiological auto-inhibited conformation. Correspondingly, the active site in the crystal structures of Arabidopsis PRORP2 does not contain metal densities and makes interactions with a neighboring molecule (10.1016/j.jmb.2015.11.025; 10.1074/jbc.M117.782078), although the protein does not require protein cofactors for activity.

The so far best available structure of a proteinaceous RNase P, that of Arabidopsis PRORP1 (ref. 22), did not include a pre-tRNA, but the 2 catalytic metal ions. The presented mtRNase P-pre-tRNA Cryo-EM structure with the modelled metal ions based on this previous structure would allow a discussion of the cleavage mechanism (phosphodiester hydrolysis), by also taking into account previous biochemical studies of proteinaceous RNase P (see discussion in 10.3390/biom6020027 and references therein).

The results section contains as a final add-on the mapping of a few of the reported disease-associated mutations in the TRMT10C and HSD17B10 genes. As poorly developed as the section presents now, it contributes little to the overall significance of the manuscript and should rather be dropped at the expense of a more thorough discussion of the above-mentioned aspects, particularly as the section does not introduce any original data and thereby appears a bit odd as a results section anyway. E.g., "mutation in T272, which is located close to the interface between MRPP1 and MRPP2 and may thus affect mtRNase P complex formation" (lines 287-8) or "Both of these residues may be required for the structural integrity of this binding groove" (lines 292-3) would have been easily testable or have previously been studied by others with conflicting result (P210S in ref. 26). The most common mutation in HSD17B10, R130C, is not addressed despite being previously hypothesized to affect the interaction with TRMT10C.

Other general issues:

The "mtRNase P field" currently suffers from the use of two different nomenclatures for the 3 subunits of the enzyme. The originally suggested MRPP1-3 names (ref. 10) were later suggested by the same group to be replaced by TRMT10C, SDR5C1 and PRORP, respectively (ref. 12). Currently, both names are found in the scientific literature. In order to approach a common nomenclature in the field I would urge switching to the latter nomenclature (TRMT10C, SDR5C1, PRORP) in this manuscript. It makes not only use of HGNC approved names, but also reflects the evolutionary relationship and primary functional significance of each subunit.

Reference to previous work appears often inappropriate or incomplete.

line 32 (ref. 2): While this is an excellent review on transcription and replication of mtDNA, it does not appropriately cover nature of mitochondrial transcripts and polycistronic arrangement of the genes.

Replace by, e.g., ref. 1 or similar.

line 37 (ref. 3): reference to the original papers that first demonstrated the concept (tRNA punctuation) and the enzymatic activities (10.1038/290470a0; 10.1074/jbc.270.21.12885; ref. 10; 10.4161/rna.8.4.15393; 10.1371/journal.pone.0019152) would be more appropriate here.

line 42 (ref. 4): Better refer to more recent reviews (e.g., 10.3390/biom6020027; 10.1007/978-3-

030-57246-4_11; 10.3390/biom6030030) than to this outdated review from before the identification of proteinaceous RNase P or the availability of any RNase P structure.

line 57 (ref. 13): Delete, as methylation was not studied in this paper.

line 61 (ref. 14): Delete; not an original finding of this study.

line 121/122 (ref. 12): Move the reference to the end of the sentence or repeat there, as it was already reported in this work that NADH does not interfere with RNase P function.

line 140 (refs. 7 and 10): add ref. 26, where the stoichiometry is probably best demonstrated.

line 172 (ref. 7): add ref. 12, where the requirement for MRPP2 was first demonstrated.

line 278: add (10.1007/978-3-030-57246-4_11) where this view was previously discussed and suggested.

line 326 (ref. 18): Inappropriate; replace by ref. 5 and (10.1093/molbev/msv187).

Nucleotide (base) positions in tRNAs are by general convention not numbered by simple counting from the 5' end, but by their canonical structural position (10.1002/wrna.103). In this way, e.g., the anticodon triplet always carries the numbers 34 to 36, and not 30 to 32. This is also true for human mitochondrial tRNAs (10.1038/s41467-020-18068-6). The numbering of the positions of mitochondrial tRNA-Tyr should be revised accordingly throughout the manuscript and all concerned figures.

Further issues in order of appearance:

line 27: "... provides the mechanism of RNA processing in human mitochondria" appears too general and overstated.

line 36: "... by the mtRNase P and Z complexes": Rephrase to "... by mtRNase P and RNase Z, ...". There is currently no solid evidence that human mitochondrial RNase Z (ELAC2) is a complex.

lines 40-6: Fungi are not a domain of life like Bacteria, Archaea or Eukaryotes. In the last mentioned, proteinaceous RNases P appear to be as common as the ribonucleoprotein forms, and the latter are not restricted to eukaryotic nuclei either, but also found in mitochondria and chloroplasts (10.1093/molbev/msv187). The sentence could be revised to "The RNase P enzymes found in most Bacteria, Archaea, and many Eukaryotes, which ..." and the reference undated as suggested above. Correspondingly, proteinaceous RNases P are more widespread in Eukarya than suggested in the following and found in nuclei and/or organelles in all possible combinations (10.1093/molbev/msv187). Finally, mammalian mtRNase P is not a "contrasting" again other form of proteinaceous RNase P, but rather homologous form with extra subunits and functions. The paragraph should be revised.

lines 86-7: I don't think that the presented data "explain the unique emergence of a three-subunit proteinaceous RNase P in mammalian mitochondria".

line 153: "C2-OH": Should be revised to "O2" or "C2-carbonyl" or "C2=O".

line 163: I was wondering whether really all the contacts with bases in the anticodon loop are specific? At least the stacking of F177 against position 35 (U31 in the manuscript) can't be, as the identity of this base varies between the different tRNAs.

line 211: add "as also suggested by previous biochemical and structural analyses for single-subunit PRORPs (10.1093/nar/gkw080; 10.1261/rna.061457; ref. 18)."

lines 272-3: U33 (in are cases C33; in the manuscript the position is labeled 29) is a conserved feature of tRNAs in general, not only human mitochondrial tRNAs.

line 305: Not all other RNase P enzymes are ribozymes; actually, proteinaceous RNases P are widespread in Eukaryotes (see also above), and a specific form was even identified in some Bacteria and Archaea (10.1007/978-3-030-57246-4_11).

lines 307-9, and 334-5: I do not see that the presented structure "suggests how the MRPP1/MRPP2 subcomplex may serve as a multi-functional platform for mitochondrial RNA processing". There are no data presented that would address this hypothesis, or "explain how this complex can act as a processing platform".

lines 345-9: The authors pickup the idea that the TRMT10C-SDR5C1 complex also supports 3'-end cleavage by RNase Z (ELAC2) from ref. 13. However, the results presented here do not contribute anything new to evaluate this hypothesis. The availability of a crystal structure of the yeast ELAC2 homolog Trz1 would have in principle allowed to model ELAC2 on the structure of the mitochondrial TRMT10C-SDR5C1-pre-tRNA complex, to reveal whether this would at all be possible without steric clashes, with the limitation that is not known how ELAC2/Trz1 binds pre-tRNA.

lines 355-9: The idea that the TRMT10C-SDR5C1 complex may function as an RNA chaperone, facilitating tRNA folding through its shape-complementarity, is intriguing, but it is not clear how this idea can be reconciled with the suggested flexibility of the NTD of TRMT10C "swinging in to form a tight grip on the tRNA". So, is it the tRNA structure that molds TRMT10C, or TRMT10C that molds the tRNA? I think, here the concepts should be better developed and clarified.

line 406: "17:3 mixture" is a rather weird and unclear description of the elution buffer. Do the authors actually mean 17 + 3 volumes? It would be better to specify the composition and concentrations.

Figure 1b: the tRNA structure should be shown in the standard orientation, with the anticodon loop at the bottom and the T loop facing to the right.

A final comment:

As a significant part of the results section actually consists of interpretation and discussion of the reported structure and the actual discussion section adds only a few additional aspects, the manuscript might benefit from merging the results and discussion sections, if journal policy allows.

Reviewer #2:

Remarks to the Author:

It is terrific to see the entire structure of the mitochondrial RNase P complex that further explains its function in vivo. This is a very compelling study that provides detailed understanding about the interactions between the MRPP1/MRPP2 complex and the MRPP3 nuclease. The authors very clearly show the requirement for the MRPP1/MRPP2 platform that is adapted to specifically accommodate the unusually structured mt-tRNAs and how this is required for MRPP3 activity, that differs from the plant PRORPs. This beautiful structure further confirms and validates the role of RNase P in mt-tRNA

processing and tRNA methylation. The authors confirm that the PPR domains are required for stabilisation of interactions instead of sequence-specific binding, similarly to POLRMT that is also different to plant mt-RNA recognition by P-type PPR proteins. Interestingly, the authors suggest that the tetrameric base formed by MRPP2 could enable binding of two MRPP1 and pre-tRNA molecules. Just out of interest, is there any indication if some substrates would be more favoured compared to others, since there are some unprocessed transcripts that are always more stable in vivo?

Overall this is an excellent study from world leaders in structural and cryo-EM biology and this is a very timely discovery that should be published.

Aleksandra Filipovska

Reviewer #3:

Remarks to the Author:

Bhatta et al. reports the first complex structure of the proteinaceous human mitochondrial RNase P bound to a pre-tRNA substrate. This long-awaited structure, solved at 3.0 Å using cryo-EM, reveals how the three protein subunits of mtRNase P coordinately recognize different structural features of a mitochondrial tRNA — its anticodon loop, elbow, variable loop, and termini. Further, the structure clearly illustrates how the dual catalytic centers are configured to catalyze the N1-methyl transfer to tRNA G9 and hydrolytic removal of pre-tRNA 5'-leader. By comparison with apo structures of individual subunits, the work further provides mechanistic insights into the alleviation of autoinhibition of the endonuclease domain, and how the MRPP1/2 subunits serve as a platform or pedestal to orchestrate multiple, sequential processing events of the mitochondrial primary transcript.

Overall the structure is of high technical quality and biological significance. The illustration is clear, minimalistic and aesthetically pleasing. Since individual structures of most components are already known, the primary novelty of this work lies in the modular, swappable architecture and myriad protein-protein and protein-pre-tRNA interfaces. One major drawback is the lack of targeted mutational analyses. However, this is largely mitigated by supporting information in the literature. Taken together, it is this reviewer's opinion that this novel structural work represents a significant advance. It rationalizes decades of biochemical, genetics, and footprinting findings and provides a framework for further work in understanding and targeting mito RNase P. Thus I recommend this work for publication in NSMB, provided that the following, mostly minor concerns can be sufficiently addressed. In addition, the length of the manuscript and references are appropriate.

Specific comments:

1. While the figures are nice looking, almost no hydrogen bonds are indicated throughout all the figures. Since we represent 3D structures as 2D projections, it is customary and necessary to indicate proposed hydrogen bonds. Otherwise, it is impossible to tell what is close to what due to the loss of depth information. I would also suggest the authors define explicitly what allowable distance range is used to propose hydrogen bonds, since including distance information for all the H-bonds would unnecessarily crowd the figures.
2. In Ex data Fig. 1a, the authors observed a second peak of the complex, which seems to contain some subunits of MRPP and increased tRNA content. Could this represent a different functional

complex? Is there a MRPP3-pre-tRNA or MRPP1/2-pre-tRNA subcomplex present here? In the EM analysis of the first peak used to solve the structure, is the tRNA stoichiometric? Did the authors observe on the grid any apo MRPP1/2/3 complex with no pre-tRNA bound?

3. Fig. 1 and text, The use of "adaptor", "adaptor helix", "adaptor loop", "adaptor domain" and later "connector helix" can be a bit confusing. I suggest the authors define this early and more clearly, and in Fig. 1c change the "adaptor" annotation to "adaptor loop", as the arrow points at the loop portion, similar to Fig. 1d.

4. Fig. 1b. The authors should define "VR" in the legend and/or the text, which is presumably "variable region" ?

5. Page 3, lines 106 and 109, the authors don't need to define NTD twice.

6. Page 4, line 149-150 and attendant Fig. 2c. The Q107-G30 base interaction is hard to see and make sense of. Is there hydrogen bonds or cation/anion-pi interaction here? This is just one example where showing hydrogen bonds, if any, would inform the nature of the interactions (see comment #1). Further, is the Q107 interaction nucleobase-specific?

7. Following comment #6, how much sequence/nucleobase selectivity does the observed MRPP1/2 interaction impose on the mito tRNA anticodon stem loop? Presumably the MRPP complex needs to bind all 22 mito pre-tRNAs of various anticodon sequences. Are the contacts observed here compatible with all these sequences, or the authors expect to see different contacts with other pre-tRNAs? For instance, must the first position of the anticodon be a purine to bind Q107? Could this protein-RNA interface have in part shaped codon/anticodon usage in the human mitochondria?

8. Page 3, line 108 and Page 4, line 156, I don't think "entangle" is accurate here, as there is no "mixing", "interweaving" or "twisting" of the structures here. Perhaps consider using "encase", "enclose", "sandwich", etc, instead.

9. Page 4, line 161. The authors discussed the widening of the ASL and DSL grooves. Is it clear whether MRPP binding causes this groove widening, or this is an inherent structural distinction between cytosolic and mitochondrial tRNAs?

10. Fig. 4a, the clashes with the pre-tRNA in faint white color are hard to see, also due to the angle of observation. If the authors want to preserve this angle for comparison with the right panel, another panel can be added in extended data to show a tRNA side view, which should clearly show the clash and the rotation.

11. Fig. 4c and associated text, page 5 line 207-9. The authors mention a "positively charged groove" but there is no visual of its electrostatic surface. Again, due to lack of H-bond notations, it is unclear which backbone is being recognized by which amino acid. Further, alpha 3 is mentioned in the text but not labeled in Fig. 4c. Is it the cyan helix in the back? Notably, Y183 appears to pack against the ribose of A51 at the apex of the T-loop, if not close enough to stack with its nucleobase. Are these within reasonable distances ($\sim 3 - 4.5 \text{ \AA}$) to interact (mostly by Van der Waals interactions)? This type of nucleobase-ribose packing interactions are frequently observed in RNA-RNA complex structures. Is Y183 a conserved aromatic residue? The manuscript can benefit from having an alignment/conservation analysis ex data figure.

12. Continuing with #11, is the characteristic structure of the pentanucleotide T-loop motif recognized? There may not be sequence specificity, but there could still be structural specificity.
13. Could the authors comment, perhaps in Discussion, on whether the mito RNase P structure possesses any anti-determinants against potentially binding or processing cytosolic pre-tRNAs? Is the canonical D-loop length/conformation and D-loop-T-loop tertiary base pairing and intercalation (involving G18 and G19 of the D-loop) in cytosolic pre-tRNAs incompatible with association with mito RNase P?
14. Page 5, line 216. The authors mention a “four-residue insertion...involved in interactions” but do not name them. They should be specified. Line 218, the authors again mention “basic residues in the nuclease domain” without naming them. These vague mentions should be specified and key contacts annotated in the figures.
15. Page 7, line 271-276. Organizational issue. Here, the authors gave a nice detailed summary of the three chief MRPP1/2 interactions. However, this seems out of place and belongs better in the preceding section on MRPP1/2 as a summary, instead of here regarding MRPP3 interactions.
16. Page 7, line 306-7. Does the structure provide insights into potential coordination or allosteric communication between the two catalytic sites? Is it known which reaction occurs first and is the sequence obligatory?
17. Page 8, line 323. This evolutionary discussion involving the “dramatic genome compaction” should be clarified. Do the authors mean that the diversity or flexibility of mito tRNA structures required multiple discrete domains (as opposed to a single domain) that can move/flex relative to each other, thus driving the evolution of a multi-domain or multi-subunit architecture as seen here in mt RNase P?
18. Page 8, line 339. The authors describe the sequence of the processing events. Is there clear experimental evidence for it? If yes citations should be provided. If no the authors should qualify the statement as a hypothesis.

Author Rebuttal to Initial comments

Reviewers' Comments:

Reviewer #1:

Remarks to the Author:

The manuscript by Bhatta et al. reports the cryo-EM structure of human mitochondrial RNase P in complex with a mitochondrial tRNA precursor. Although the study does not include any accompanying biochemical analyses to substantiate its mechanistic implications/conclusions, the mere structure on its own represents a major scientific achievement in the field. It represents the first structure of a proteinaceous RNase P in complex with its pre-tRNA substrate and at the same time also the first structure of a TRM10-type methyltransferase with its tRNA substrate. Together with previously published crystallographic studies of individual proteins or protein domains, as well as biochemical studies, the cryo-EM structure of the holoenzyme-substrate complex allows a new level of insight and will certainly inspire the design of future studies testing its mechanistic and evolutionary implications. While this new structure thus in my eyes deserves being published, the current report falls short

in appropriately putting the data in the context of previously published work and makes mechanistic and evolutionary claims that are not (sufficiently) supported by the available structural information.

We thank the reviewer for his or her comments, which we feel have greatly improved the manuscript.

References in the following are either indicated by the reference number used in the manuscript or, if new, by their DOI.

The authors conclude that (lines 188-9) “the structure of the mtRNase P complex provides the mechanistic basis for N1-methylation at position 9 of tRNAs.” However, although the structure nicely shows the expected rotation of the purine of position 9 out of the tRNA core and some of its interactions within the active site, the authors don’t even attempt to address the actual mechanism of methylation. Notably, the mechanism of methylation by the TRM10 class of methyltransferases has been elusive and remains unclear despite active-site mutagenesis studies and crystal structures of several TRM10 methyltransferases, all of them without a tRNA substrate, however. Given the latter limitation of previous studies, the presented structure could indeed be expected to provide some insight into the catalytic mechanism, despite the lack of the SAM cofactor, yet the issue is not actually touched on in the manuscript.

We have added a paragraph that discusses our structural data in the context of the proposed mechanisms of TRM10 methyltransferases in the discussion (lines 342-357). In addition to the previously modeled pre-catalytic state with G in the active site, we have now also included a model of the pre-catalytic state with A in the active site (Extended Data Figure 5d), which we discuss in the results section (lines 190-193) and the discussion section (lines 352-354).

Moreover, a comparison of the MRPP1 (TRMT10C) structure to the 2 previously published full-length TRM10 structures (10.1093/nar/gkv1369; 10.1261/rna.064345.117) could be informative. Other than suggested by the misleading introduction of TRMT10C’s gross structure (lines 48-51), all TRM10 enzymes appear to have a homologous N-terminal domain (NTD) in addition to their SPOUT methyltransferase domain. This NTD appears thus not specifically required for the pre-tRNA-processing role of TRMT10C in the mtRNase P complex, but apparently for the interaction of TRMT10C with the tRNA (as also demonstrated by the presented structure). A comparison of the available structures could indicate whether other TRM10 enzymes possibly interact in a similar way with their tRNA substrates. How do their NTD structures compare? Do they all contain a “connector” helix lined by basic residues, like TRMT10C? Are the connections between the SPOUT domains and the NTDs of other TRM10 enzymes flexible enough to encase a tRNA in between them in a similar way? Such a comparison could finally also provide some insight in the unique dependence of TRMT10C on the MRPP2 tetramer for methylation, a key question that is not sufficiently addressed by the current analysis of the mtRNase P structure.

We have extended the introduction of MRPP1/TRMT10C to put it into context of other TRMT10 enzymes (lines 48-58). In addition, we have added a structural and sequence comparison of MRPP1/TRMT10C with the previously published full-length TRM10 enzymes (Extended Data Figure 6) and added a paragraph on the comparison to the results section on the MRPP1/MRPP2 complex structure (lines 168-174).

The discussion of the recruitment and activation of the nuclease subunit, a supposedly dynamic process, is based on a comparison to 2 previously reported crystal structures of MRPP3 fragments and entirely follows the scenario suggested in those papers (refs. 14 and 15). However, others have more recently raised substantial criticism in this interpretation (10.3390/biom6020027; 10.1007/978-3-030-57246-4_11). As the new structure does not resolve the concerns raised and no additional biochemical or structural data are presented that would address this point, I think

the discussion should more critically address the issue of the “activation” of the nuclease subunit. E.g., sentences like “In the previous crystal structures of apo-MRPP3, the enzyme adopted an auto-inhibited conformation ...” (line 195) appear misleading, as the previously crystallized fragments were enzymatically inactive even in the presence of the MRPP1/MRPP2 (thus neither “apo-MRPP3” nor “enzyme” appear appropriate in this context). In fact, the cryo-EM structure now demonstrates that those MRPP3 fragments were obviously inactive due to the deletion of the N-terminal part of the PPR domain, which is shown to be involved in the interaction of MRPP3 with the pre-tRNA and MRPP1; similar N-terminal deletions of single-subunit PRORPs were likewise inactive, by the way. “This explains why this region of MRPP3 is critical for activity although it was not resolved in the previous MRPP3 crystal structure” (lines 205-7) is thus also misleading, because the addressed region was simply not included in the crystallized MRPP3 fragments. As discussed previously (10.3390/biom6020027; 10.1007/978-3-030-57246-4_11) it still appears equally reasonable that the distortion of the active site is related to the N-terminal deletion, rather than representing a physiological auto-inhibited conformation. Correspondingly, the active site in the crystal structures of Arabidopsis PRORP2 does not contain metal densities and makes interactions with a neighboring molecule (10.1016/j.jmb.2015.11.025; 10.1074/jbc.M117.782078), although the protein does not require protein cofactors for activity.

We have rephrased the results section for MRPP3 and removed the discussion about the potential functional implications of the ‘closed’ conformation of MRPP3 in that section. Instead, we have added a paragraph to the discussion in which we emphasize the limitations of previous structural studies (and cite the suggested papers) and suggest a potential model for activation of MRPP3 if apo-MRPP3 indeed adopts an auto-inhibited state (lines 318-329).

The so far best available structure of a proteinaceous RNase P, that of Arabidopsis PRORP1 (ref. 22), did not include a pre-tRNA, but the 2 catalytic metal ions. The presented mtRNase P-pre-tRNA Cryo-EM structure with the modelled metal ions based on this previous structure would allow a discussion of the cleavage mechanism (phosphodiester hydrolysis), by also taking into account previous biochemical studies of proteinaceous RNase P (see discussion in 10.3390/biom6020027 and references therein).

We have added a paragraph discussing our structure in the context of previously proposed cleavage mechanisms for ribozyme-based and PRORP enzymes to the discussion (lines 330-341).

The results section contains as a final add-on the mapping of a few of the reported disease-associated mutations in the TRMT10C and HSD17B10 genes. As poorly developed as the section presents now, it contributes little to the overall significance of the manuscript and should rather be dropped at the expense of a more thorough discussion of the above-mentioned aspects, particularly as the section does not introduce any original data and thereby appears a bit odd as a results section anyway. E.g., “mutation in T272, which is located close to the interface between MRPP1 and MRPP2 and may thus affect mtRNase P complex formation” (lines 287-8) or “Both of these residues may be required for the structural integrity of this binding groove” (lines 292-3) would have been easily testable or have previously been studied by others with conflicting result (P210S in ref. 26). The most common mutation in HSD17B10, R130C, is not addressed despite being previously hypothesized to affect the interaction with TRMT10C.

As suggested by the reviewer, we have removed this section and have limited the discussion of disease-associated mutations to mentioning R181 substitutions in the results section describing the interaction of R181 with the anticodon loop. As we believe the figure mapping known mutations may nevertheless be useful for some readers, we suggest to keep it as Extended Data Figure 4.

Other general issues:

The “mtRNase P field” currently suffers from the use of two different nomenclatures for the 3 subunits of the enzyme. The originally suggested MRPP1-3 names (ref. 10) were later suggested by the same group to be replaced by TRMT10C, SDR5C1 and PRORP, respectively (ref. 12). Currently, both names are found in the scientific literature. In order to approach a common nomenclature in the field I would urge switching to the latter nomenclature (TRMT10C, SDR5C1, PRORP) in this manuscript. It makes not only use of HGNC approved names, but also reflects the evolutionary relationship and primary functional significance of each subunit.

We agree with the reviewer that the names TRMT10C, SDR5C1 and PRORP are concise and consistent with standardized naming. However, we feel that this nomenclature may be confusing to the broad non-expert readership of this paper, and would thus favor MRPP1-3 throughout the text. However, we have clearly introduced the nomenclature in the introduction and also include both nomenclatures in Figure 1. We think this is a good compromise between ensuring consistent naming in the literature and readability of the manuscript, and hope the reviewer agrees.

Reference to previous work appears often inappropriate or incomplete.

line 32 (ref. 2): While this is an excellent review on transcription and replication of mtDNA, it does not appropriately cover nature of mitochondrial transcripts and polycistronic arrangement of the genes. Replace by, e.g., ref. 1 or similar.

We have replaced the reference with reference 1.

line 37 (ref. 3): reference to the original papers that first demonstrated the concept (tRNA punctuation) and the enzymatic activities (10.1038/290470a0; 10.1074/jbc.270.21.12885; ref. 10; 10.4161/rna.8.4.15393; 10.1371/journal.pone.0019152) would be more appropriate here.

We have included these references.

line 42 (ref. 4): Better refer to more recent reviews (e.g., 10.3390/biom6020027; 10.1007/978-3-030-57246-4_11; 10.3390/biom6030030) than to this outdated review from before the identification of proteinaceous RNase P or the availability of any RNase P structure.

We have replaced this reference with 10.3390/biom6020027 & 10.1007/978-3-030-57246-4_11.

line 57 (ref. 13): Delete, as methylation was not studied in this paper.

We have removed this reference.

line 61 (ref. 14): Delete; not an original finding of this study.

We have removed this reference.

line 121/122 (ref. 12): Move the reference to the end of the sentence or repeat there, as it was already reported in this work that NADH does not interfere with RNase P function.

We have removed this sentence altogether due to length restrictions.

line 140 (refs. 7 and 10): add ref. 26, where the stoichiometry is probably best demonstrated.

We have added ref.26 as suggested.

line 172 (ref. 7): add ref. 12, where the requirement for MRPP2 was first demonstrated.

Ref. 12 has been added.

line 278: add (10.1007/978-3-030-57246-4_11) where this view was previously discussed and suggested.

We have added this reference.

line 326 (ref. 18): Inappropriate; replace by ref. 5 and (10.1093/molbev/msv187).

We have removed this sentence altogether due to length restrictions.

Nucleotide (base) positions in tRNAs are by general convention not numbered by simple counting from the 5' end, but by their canonical structural position (10.1002/wrna.103). In this way, e.g., the anticodon triplet always carries the numbers 34 to 36, and not 30 to 32. This is also true for human mitochondrial tRNAs (10.1038/s41467-020-18068-6). The numbering of the positions of mitochondrial tRNA-Tyr should be revised accordingly throughout the manuscript and all concerned figures.

We thank the reviewer for pointing this out and agree that numbering should follow the general convention. Unfortunately, structural models in PDB/mmCIF format require consecutive numbering of residues for proper linkage and display, and we therefore cannot adopt this numbering in the models. We have therefore revised the nucleotide position labels in figures and text throughout the manuscript to contain the numbering according to our construct followed by the according canonical tRNA numbering in parenthesis. Furthermore, all references to general tRNA positions have been revised to follow canonical numbering. This way, the reader can both deduce the canonical numbering while reading the manuscript and also easily cross-reference with the numbering in the deposited structural model.

Further issues in order of appearance:

line 27: "... provides the mechanism of RNA processing in human mitochondria" appears too general and overstated.

We have changed this to "...provides the molecular basis for the first step of RNA processing in human mitochondria."

line 36: "... by the mtRNase P and Z complexes": Rephrase to "... by mtRNase P and RNase Z, ...". There is currently no solid evidence that human mitochondrial RNase Z (ELAC2) is a complex.

We have rephrased as suggested.

lines 40-6: Fungi are not a domain of life like Bacteria, Archaea or Eukaryotes. In the last mentioned, proteinaceous RNases P appear to be as common as the ribonucleoprotein forms, and the latter are not restricted to eukaryotic nuclei either, but also found in mitochondria and chloroplasts (10.1093/molbev/msv187). The sentence could be revised to "The RNase P enzymes found in most Bacteria, Archaea, and many Eukaryotes, which ..." and the reference undated as suggested above. Correspondingly, proteinaceous RNases P are more widespread in Eukarya than suggested in the following and found in nuclei and/or organelles in all possible combinations (10.1093/molbev/msv187). Finally, mammalian mtRNase P is not a "contrasting" again other form of proteinaceous RNase P, but rather homologous form with extra subunits and functions. The paragraph should be revised.

We have rewritten this paragraph and included the suggested reference.

lines 86-7: I don't think that the presented data "explain the unique emergence of a three-subunit proteinaceous RNase P in mammalian mitochondria".

We have rephrased this sentence to "rationalize the unique emergence of a three-subunit proteinaceous RNase P in mammalian mitochondria", because the structural data provide insights into the role of the additional factors in substrate binding and recognition.

line 153: "C2-OH": Should be revised to "O2" or "C2-carbonyl" or "C2=O".

We have changed this to "C2-carbonyl".

line 163: I was wondering whether really all the contacts with bases in the anticodon loop are specific? At least the stacking of F177 against position 35 (U31 in the manuscript) can't be, as the identity of this base varies between the different tRNAs.

We agree that this statement was not accurate, and have rephrased it to “In summary, the MRPP1/MRPP2 subcomplex contacts all four arms of the pre-tRNA and interacts with the substrate through both non-specific and specific interactions.”

line 211: add “as also suggested by previous biochemical and structural analyses for single-subunit PRORPs (10.1093/nar/gkw080; 10.1261/rna.061457; ref. 18).”

The sentence has been added (slightly shortened) with these references, as suggested.

lines 272-3: U33 (in are cases C33; in the manuscript the position is labeled 29) is a conserved feature of tRNAs in general, not only human mitochondrial tRNAs.

We thank the reviewer for pointing this out. As described above, we have changed the annotation of tRNA residues to include both specific construct and canonical numbering. We have moved this statement to the section describing MRPP1/MRPP2 interactions with the pre-tRNA, and rephrased it to reflect that U or C are conserved at this position in all tRNAs (lines 154-157).

line 305: Not all other RNase P enzymes are ribozymes; actually, proteinaceous RNases P are widespread in Eukaryotes (see also above), and a specific form was even identified in some Bacteria and Archaea (10.1007/978-3-030-57246-4_11).

We thank the reviewer for pointing this out. We have removed the respective sentence as it is redundant with the introduction and replaced it with: “The structure reveals how pre-tRNA is recognized, cleaved and methylated and suggests an explanation for the emergence of the evolutionarily unique trimeric proteinaceous mtRNase P.” (lines 304-305).

lines 307-9, and 334-5: I do not see that the presented structure “suggests how the MRPP1/MRPP2 subcomplex may serve as a multi-functional platform for mitochondrial RNA processing”. There are no data presented that would address this hypothesis, or “explain how this complex can act as a processing platform”.

While it is true that we do not present data that would confirm or disprove the hypothesis that MRPP1/2 can act as a processing platform, the structural data does provide a molecular model of how a MRPP1/2 subcomplex would likely interact with the pre-tRNA in the context of such a platform complex. In order to clarify this, we have removed the respective sentence in the former lines 307-309, and expanded the discussion section by explaining that the structure of the MRPP1/2-pre-tRNA complex would be consistent with such a model (lines 377-382).

lines 345-9: The authors pickup the idea that the TRMT10C-SDR5C1 complex also supports 3'-end cleavage by RNase Z (ELAC2) from ref. 13. However, the results presented here do not contribute anything new to evaluate this hypothesis. The availability of a crystal structure of the yeast ELAC2 homolog Trz1 would have in principle allowed to model ELAC2 on the structure of the mitochondrial TRMT10C-SDR5C1-pre-tRNA complex, to reveal whether this would at all be possible without steric clashes, with the limitation that is not known how ELAC2/Trz1 binds pre-tRNA.

We agree with the reviewer that such modeling would be of interest. Consequently, we attempted to generate a model of a putative MRPP1/2-pre-tRNA-ELAC2 complex in preparation of this manuscript. However, as the reviewer points out, there is no structural data on ELAC2 or its interaction with the pre-tRNA, and we thus felt such a model would be too speculative to be included in the manuscript. We have clarified in the discussion that no structural data on ELAC2 is available, and it is thus not clear what the architecture of a 3' processing complex would be (lines 381-385).

lines 355-9: The idea that the TRMT10C-SDR5C1 complex may function as an RNA chaperone, facilitating tRNA folding through its shape-complementarity, is intriguing, but it is not clear how this idea can be reconciled with the suggested flexibility of the NTD of TRMT10C "swinging in to form a tight grip on the tRNA". So, is it the tRNA structure that molds TRMT10C, or TRMT10C that molds the tRNA? I think, here the concepts should be better developed and clarified.

We have rewritten this part of the discussion to clarify our chaperone model (lines 358-367).

line 406: "17:3 mixture" is a rather weird and unclear description of the elution buffer. Do the authors actually mean 17 + 3 volumes? It would be better to specify the composition and concentrations.

As suggested, the "17:3 mixture of lysis buffer and 2 M Imidazole pH 8.0" has been replaced with final composition and concentrations of the elution buffer.

Figure 1b: the tRNA structure should be shown in the standard orientation, with the anticodon loop at the bottom and the T loop facing to the right.

As suggested, the tRNA orientation has been changed such that the anticodon loop faces towards the bottom and the T loop towards the right.

A final comment:

As a significant part of the results section actually consists of interpretation and discussion of the reported structure and the actual discussion section adds only a few additional aspects, the manuscript might benefit from merging the results and discussion sections, if journal policy allows.

Journal policy requires a discussion section.

Reviewer #2:

Remarks to the Author:

It is terrific to see the entire structure of the mitochondrial RNase P complex that further explains its function in vivo. This is a very compelling study that provides detailed understanding about the interactions between the MRPP1/MRPP2 complex and the MRPP3 nuclease. The authors very clearly show the requirement for the MRPP1/MRPP2 platform that is adapted to specifically accommodate the unusually structured mt-tRNAs and how this is required for MRPP3 activity, that differs from the plant PRORPs. This beautiful structure further confirms and

validates the role of RNase P in mt-tRNA processing and tRNA methylation. The authors confirm that the PPR domains are required for stabilisation of interactions instead of sequence-specific binding, similarly to POLRMT that is also different to plant mt-RNA recognition by P-type PPR proteins. Interestingly, the authors suggest that the tetrameric base formed by MRPP2 could enable binding of two MRPP1 and pre-tRNA molecules.

Just out of interest, is there any indication if some substrates would be more favoured compared to others, since there are some unprocessed transcripts that are always more stable in vivo?

While mitochondrial tRNAs are remarkably variable in sequence composition and lengths, the tRNA elements recognized by the mtRNase P appear conserved among most tRNAs processed by mtRNase P complex. With the exception of mt-tRNA-ser(UCN), which lacks the position 9 nucleotide, and mt-tRNA-ser(AGY), which lacks the entire D arm and is not processed by mtRNase P, we expect all mitochondrial pre-tRNAs to interact with mtRNase P in a similar manner, as long as they adopt a tRNA-like fold. It is possible that the relative affinity of mtRNase P for different pre-tRNAs may differ by virtue of their sequences or stability of their tRNA-like fold. However, it is also likely that the differential stability of mitochondrial precursor transcripts observed in vivo is due to effects of other mitochondrial factors implicated in regulating mitochondrial RNA stability or due to the structural constraints imposed on the tRNA by the flanking 5' and 3' sequences, which may affect its processing and/or stability. At present, we are unable to disentangle these possibilities.

Overall this is an excellent study from world leaders in structural and cryo-EM biology and this is a very timely discovery that should be published.

Aleksandra Filipovska

We thank the reviewer for her kind words and evaluation of our work.

Reviewer #3:

Remarks to the Author:

Bhatta et al. reports the first complex structure of the proteinaceous human mitochondrial RNase P bound to a pre-tRNA substrate. This long-awaited structure, solved at 3.0 Å using cryo-EM, reveals how the three protein subunits of mtRNase P coordinately recognize different structural features of a mitochondrial tRNA — its anticodon loop, elbow, variable loop, and termini. Further, the structure clearly illustrates how the dual catalytic centers are configured to catalyze the N1-methyl transfer to tRNA G9 and hydrolytic removal of pre-tRNA 5'-leader. By comparison with apo structures of individual subunits, the work further provides mechanistic insights into the alleviation of autoinhibition of the endonuclease domain, and how the MRPP1/2 subunits serve as a platform or pedestal to orchestrate multiple, sequential processing events of the mitochondrial primary transcript.

Overall the structure is of high technical quality and biological significance. The illustration is clear, minimalistic and aesthetically pleasing. Since individual structures of most components are already known, the primary novelty of this work lies in the modular, swappable architecture and myriad protein-protein and protein-pre-tRNA interfaces. One major drawback is the lack of targeted mutational analyses. However, this is largely mitigated by supporting information in the literature. Taken together, it is this reviewer's opinion that this novel structural work represents a significant advance. It rationalizes decades of biochemical, genetics, and footprinting findings and provides a framework for further work in understanding and targeting mito RNase P. Thus I recommend this work for publication

in NSMB, provided that the following, mostly minor concerns can be sufficiently addressed. In addition, the length of the manuscript and references are appropriate.

We thank the reviewer for his or her comments, which we have addressed as outlined below.

Specific comments:

1. While the figures are nice looking, almost no hydrogen bonds are indicated throughout all the figures. Since we represent 3D structures as 2D projections, it is customary and necessary to indicate proposed hydrogen bonds. Otherwise, it is impossible to tell what is close to what due to the loss of depth information. I would also suggest the authors define explicitly what allowable distance range is used to propose hydrogen bonds, since including distance information for all the H-bonds would unnecessarily crowd the figures.

As suggested, we have updated the figures to show potential hydrogen bonds where contextually appropriate. In addition, we have indicated stacking interactions where appropriate. We have described in the methods how potential hydrogen bonds were determined.

2. In Ex data Fig. 1a, the authors observed a second peak of the complex, which seems to contain some subunits of MRPP and increased tRNA content. Could this represent a different functional complex? Is there a MRPP3-pre-tRNA or MRPP1/2-pre-tRNA subcomplex present here?

Based on analytical gel filtration experiments we performed as preliminary data (not included in the manuscript), the second peak observed likely corresponds to excess pre-tRNA along with free MRPP3 and MRPP1/2. MRPP1/2 and pre-tRNA can indeed form a complex independent of MRPP3 (as previously shown in references 15, 17 and 26) but the elution volume of this complex is close to that of the full mtRNase P complex. Free MRPP3 has been reported to have only low affinity for pre-tRNAs ($K_D \sim 1.8\text{-}12 \mu\text{M}$, K_D of MRPP1/2-pre-tRNA complex $\sim 40 \text{ nM}$) in absence of MRPP1/2 (Karasik et al. 2018, biorxiv; Liu et al. 2019, biorxiv). Although the existence of a weak complex between MRPP3 and pre-tRNA in the second peak cannot be precluded, we currently lack any indication for functional relevance of such a complex.

In the EM analysis of the first peak used to solve the structure, is the tRNA stoichiometric?

During classification of our particle data set, we selected for particles which represent the full complex consisting of MRPP1, MRPP2, pre-tRNA and MRPP3 and effectively sort out particles that do not contain pre-tRNA. Consequently, in the reconstruction from the final particle set described in this manuscript, we have no indication that the RNA is sub-stoichiometric compared to any of the other components (which would be indicated by weaker density, for example). During classification, we did not observe a significant number of particles that lack pre-tRNA. We did observe particles lacking MRPP3, which represent the MRPP1/2-pre-tRNA complex. Notably, preliminary studies conducted by us in preparation of this manuscript showed that the structure of the MRPP1/2-pre-tRNA complex is identical to its structure in the context of mtRNase P (with MRPP3).

Additionally, as also described in the text, we did observe weaker densities for a second copy of MRPP1 bound to MRPP2 in two opposite orientations as well as for a second copy of pre-tRNA in the reconstruction from the final particle set (Extended data figure 3). The weaker density for the second copy of MRPP1 and pre-tRNA indicate that

these may be present sub-stoichiometrically, and the sample thus likely contains a mixture of 4:1:1:1 and 4:2:2:2 complexes. Whether this represents the physiological occurrence of this complex or is a result of sample preparation is not entirely clear. The blotting step in grid preparation for cryo-EM sometimes results in subunit dissociation in multi-subunit complexes (Taylor and Glaeser 2008, J. Struct. Biol.; doi: 10.1016/j.jsb.2008.06.004). Thus, dissociation may be the reason why we were not able to observe a 4:2:2:2 stoichiometry for the majority of particles in our EM analysis.

Did the authors observe on the grid any apo MRPP1/2/3 complex with no pre-tRNA bound?

We did not observe apo MRPP1/2/3 complexes without pre-tRNA in our EM analysis. This is consistent with our own preliminary data that shows that MRPP1/2 and MRPP3 do not form a stable complex in the absence of pre-tRNA in gel filtration experiments (data not shown in manuscript).

3. Fig. 1 and text, The use of “adaptor”, “adaptor helix”, “adaptor loop”, “adaptor domain” and later “connector helix” can be a bit confusing. I suggest the authors define this early and more clearly, and in Fig. 1c change the “adaptor” annotation to “adaptor loop”, as the arrow points at the loop portion, similar to Fig. 1d.

As suggested, we have revised the text accordingly (removed “adaptor domain”) and modified Fig. 1 such that “adaptor helix” and “adaptor loop” are labeled in figure 1c as in figure 1d.

4. Fig. 1b. The authors should define “VR” in the legend and/or the text, which is presumably “variable region” ?

As suggested, we have defined VR in figure 1b as “variable region” in the corresponding figure legend.

5. Page 3, lines 106 and 109, the authors don’t need to define NTD twice.

We thank the reviewer for pointing this out, and have removed the second definition.

6. Page 4, line 149-150 and attendant Fig. 2c. The Q107-G30 base interaction is hard to see and make sense of. Is there hydrogen bonds or cation/anion-pi interaction here? This is just one example where showing hydrogen bonds, if any, would inform the nature of the interactions (see comment #1). Further, is the Q107 interaction nucleobase-specific?

We have indicated the interactions in the figure, as suggested, and re-phrased the text to include also potential interactions with other sidechains (S98 and K105). We have also included a statement that these interactions are likely not base-specific, as the position is not conserved as G (lines 149-150).

7. Following comment #6, how much sequence/nucleobase selectivity does the observed MRPP1/2 interaction impose on the mito tRNA anticodon stem loop? Presumably the MRPP complex needs to bind all 22 mito pre-tRNAs of various anticodon sequences. Are the contacts observed here compatible with all these sequences, or the authors

expect to see different contacts with other pre-tRNAs? For instance, must the first position of the anticodon be a purine to bind Q107? Could this protein-RNA interface have in part shaped codon/anticodon usage in the human mitochondria?

The only interaction that we determine to be unambiguously specific is the one between R181 and U29, which is specific for pyrimidines, as this position is conserved as U or C in all tRNAs. We have added a sentence to the results section explicitly stating this (lines 154-157). The only other base for which we could potentially envision specific interactions is G30, but this position is not conserved as G. We have added a sentence stating this (lines 149-150). The observed contacts are therefore compatible with all mitochondrial tRNAs, and probably even in principle allow binding of cytosolic tRNAs, as the mtRNase P complex has been shown to have partial cleavage activity on cytosolic or bacterial tRNAs (we have added this information in lines 286-288). The structure therefore suggests that the interface has evolved to accommodate a large sequence space in the anticodon loop, but recognizes a highly specific feature of tRNAs – the pyrimidine in position U29. We therefore see no indication that the RNase P – tRNA interaction could have influenced codon/anticodon usage in human mitochondria.

8. Page 3, line 108 and Page 4, line 156, I don't think "entangle" is accurate here, as there is no "mixing", "interweaving" or "twisting" of the structures here. Perhaps consider using "encase", "enclose", "sandwich", etc, instead.

We have rephrased to "encases".

9. Page 4, line 161. The authors discussed the widening of the ASL and DSL grooves. Is it clear whether MRPP binding causes this groove widening, or this is an inherent structural distinction between cytosolic and mitochondrial tRNAs?

All previous structural information on mammalian mitochondrial tRNAs is derived from mt-tRNAs in complex with mitochondrial ribosomes. Compared to these, the groove between anticodon arm (ASL) and D arm (DSL) in mt-tRNA^{Tyr} in the mtRNase P complex appears to be wider.

We believe it is unlikely that wider ASL-DSL groove is an inherent structural feature of all mitochondrial tRNAs (mt-tRNAs), because sequences of some mt-tRNAs, like tRNA^{Asn}, tRNA^{Leu(UUR)} or tRNA^{Gln}, resemble canonical (cytosolic) tRNAs and are expected to adopt canonical tRNA structure (ref. 25). MtRNase P has also been shown to be able to catalyze 5' cleavage of cytosolic or bacterial pre-tRNAs (refs 17, 39), suggesting that the widened ASL-DSL groove is not required for mt-tRNA binding or recognition by mtRNase P. Instead, the structure suggests that the widening is a result of the extensive interactions of the tRNA acceptor arm with the MRPP1 methyltransferase and N-terminal domains. Therefore, it appears likely that the mtRNase P complex induces and stabilize the ASL-DSL groove widening in tRNAs. However, since we have no further experimental evidence to substantiate this, we have explicitly stated both possibilities in the text (lines 294-295).

As we believe the distorted tRNA structure is a highly interesting and somewhat surprising feature of the mtRNase P structure, we have decided to make the figure showing this a main text figure (Figure 3).

10. Fig. 4a, the clashes with the pre-tRNA in faint white color are hard to see, also due to the angle of observation. If the authors want to preserve this angle for comparison with the right panel, another panel can be added in extended data to show a tRNA side view, which should clearly show the clash and the rotation.

As suggested, we have added an extended data figure (Extended Data 7a) which emphasizes the clashes between pre-tRNA and the MRPP3 nuclease domain. In addition, we have changed the color of the pre-tRNA in figure 4a (left) to a darker shade of grey for clarity.

11. Fig. 4c and associated text, page 5 line 207-9. The authors mention a “positively charged groove” but there is no visual of its electrostatic surface. Again, due to lack of H-bond notations, it is unclear which backbone is being recognized by which amino acid. Further, alpha 3 is mentioned in the text but not labeled in Fig. 4c. Is it the cyan helix in the back?

As suggested, we have added electrostatic surface potential representations and probable hydrogen bonds between the MRPP3 PPR domain and pre-tRNA T-arm to figure 4. We have additionally referenced figure 2d, which shows alpha 3 of MRPP1 and its lining with positively charged residues.

Notably, Y183 appears to pack against the ribose of A51 at the apex of the T-loop, if not close enough to stack with its nucleobase. Are these within reasonable distances (~3 - 4.5 Å) to interact (mostly by Van der Waals interactions)? This type of nucleobase-ribose packing interactions are frequently observed in RNA-RNA complex structures. Is Y183 a conserved aromatic residue? The manuscript can benefit from having an alignment/conservation analysis ex data figure.

As noted by the reviewer, the aromatic ring of Y183 and the ribose of A51 are within reasonable distance (3.1 Å between O4' of ribose and Y183 ring center) to interact. We have now indicated this interaction in figure 4c and 5d, and added a sentence describing it in the MRPP3 section (lines 228-229). Furthermore, we have added a multiple sequence alignment of PPR domains among PRORPs (Extended Data Figure 8) which shows conservation of Y183 among PRORPs across species. Notably, the corresponding residue in PRORPs is involved in interactions with the conserved C56-A19 base pair, as discussed in the manuscript section comparing mtRNase P to other RNase P enzymes (lines 278-283).

12. Continuing with #11, is the characteristic structure of the pentanucleotide T-loop motif recognized? There may not be sequence specificity, but there could still be structural specificity.

In canonical tRNAs, the T-loop generally has a conserved length of 7 nucleotides. Herein, the pentanucleotide T-loop motif comprises of U-A Reverse-Hoogsteen base-pairing between nucleobases at positions 1 and 5 in the T-loop, stabilized by stacking with bases at positions 2 and 4 respectively, to form a compact U-turn-like structure. Therefore, the formation of this motif is dependent on the length as well as sequence conservation in T loops. The T-loop of the mt-tRNA-Tyr used in this study comprises only of 5 nucleotides and, in our structure, mt-tRNA-Tyr therefore does not possess the characteristic pentanucleotide structure described above. We cannot rule out that additional interactions may be established between mtRNase P and other pre-tRNA substrates that more closely resemble “canonical” tRNAs, and we have emphasized this in text (lines 285-288). However, in mitochondrial tRNAs, the length (ranging from 3 to 9 nts) and sequence of T loops is highly variable, and mtRNase P processes all but one of these tRNAs. Based on this and the structure of mtRNase P in complex with pre-tRNA-Tyr, we therefore conclude that the pentaloop is unlikely to be a strong specific determinant of substrate recognition.

13. Could the authors comment, perhaps in Discussion, on whether the mito RNase P structure possesses any anti-

determinants against potentially binding or processing cytosolic pre-tRNAs? Is the canonical D-loop length/conformation and D-loop-T-loop tertiary base pairing and intercalation (involving G18 and G19 of the D-loop) in cytosolic pre-tRNAs incompatible with association with mito RNase P?

MtRNase P has in fact been shown to be able to cleave cytosolic as well as bacterial pre-tRNAs (refs 17,39) and some mitochondrial tRNAs (tRNA-Asn, tRNA-Leu(UUR) or tRNA-Gln) are expected to contain D-T loop interactions. From the structure, we see no indications that mtRNase P would anti-discriminate against cytosolic pre-tRNAs. The tip of the D-loop and MRPP3 regions close to it are poorly resolved in our structure, which indicates a lack of ordered interactions. However, as we note in the text (lines 283-287), pre-tRNAs with longer D-loops or canonical t-RNAs with D-T loop interactions may lead to additional interactions in these regions. We have added a sentence explicitly stating that mtRNase P can likely bind also cytosolic pre-tRNAs, as has been shown in vitro (lines 287-289).

14. Page 5, line 216. The authors mention a “four-residue insertion...involved in interactions” but do not name them. They should be specified. Line 218, the authors again mention “basic residues in the nuclease domain” without naming them. These vague mentions should be specified and key contacts annotated in the figures.

We have added the corresponding residue ranges or specified them explicitly. We have chosen to remove the statement on the insertion, as it is conserved in mammals and absent in PRORP1, but an insertion with different sequence exists in PRORP2 and PRORP3.

15. Page 7, line 271-276. Organizational issue. Here, the authors gave a nice detailed summary of the three chief MRPP1/2 interactions. However, this seems out of place and belongs better in the preceding section on MRPP1/2 as a summary, instead of here regarding MRPP3 interactions.

As suggested by the reviewer, we considered moving this paragraph to the part on MRPP1/2 or to the discussion. However, after careful consideration, we believe this summary makes most sense at the end of the paragraph on the comparison to other RNase P enzymes, because it recapitulates the major RNA interactions that compensate for the absence of specific interactions between MRPP3 and the pre-tRNA that would correspond to those formed by other RNase P enzymes. However, we have slightly rephrased this section (now lines 289-298) and hope the reviewer agrees that it results in the best flow of the text.

16. Page 7, line 306-7. Does the structure provide insights into potential coordination or allosteric communication between the two catalytic sites? Is it known which reaction occurs first and is the sequence obligatory?

The structure provides no indications for a coordination or communication between the two active sites. This is consistent with previous reports that suggest that methylation and 5' cleavage are independent (reference 17). We have now explicitly stated this in the discussion (lines 374-376).

17. Page 8, line 323. This evolutionary discussion involving the “dramatic genome compaction” should be clarified. Do the authors mean that the diversity or flexibility of mito tRNA structures required multiple discrete domains (as opposed to a single domain) that can move/flex relative to each other, thus driving the evolution of a multi-domain or multi-subunit architecture as seen here in mt RNase P?

We have rephrased this part of the discussion to clarify this (lines 310-312).

18. Page 8, line 339. The authors describe the sequence of the processing events. Is there clear experimental evidence for it? If yes citations should be provided. If no the authors should qualify the statement as a hypothesis.

We have added a sentence clarifying that the order of events is not known (lines 374-377) and have modified the starting sentence of this paragraph to explicitly state that this is a model (lines 367-369)

Additional changes to the manuscript:

- *After careful consideration, we have decided to carry out final refinement against a composite map of Map 1 and Map 2, as described in Methods, as this led to a better final model.*
- *The view in Figure 3 and Extended Data Figure 5 was slightly modified to clarify all points discussed in the text.*
- *We have re-written parts of the manuscript to adhere to length suggestions by the journal.*
- *We have updated Extended Data Table 1 to reflect the statistics of the final model submitted to the PDB.*

Decision Letter, first revision:

14th Jun 2021

Dear Hauke,

Thank you again for submitting your revised manuscript "Structural basis of RNA processing by human mitochondrial RNase P" (NSMB-A44742A). It has now been seen by two of the original referees and their comments are below. The reviewers find that the paper has improved in revision, and we'll be happy in principle to publish it in Nature Structural & Molecular Biology, pending further textual revisions to satisfy the referees' final requests and to comply with our editorial and formatting guidelines.

We are now performing detailed checks on your paper and will send you a checklist detailing our editorial and formatting requirements in about a week. Please do not upload the final materials until you receive this additional information from us.

** To facilitate our work at this stage, we would appreciate if you could send us the main text as a word file. Please make sure to copy the NSMB account (cc'ed above).

Thank you again for your interest in Nature Structural & Molecular Biology Please do not hesitate to

contact me if you have any questions.

With kind regards,
Anke

Anke Sparmann, PhD
Senior Editor
Nature Structural and Molecular Biology
ORCID 0000-0001-7695-2049

Author Rebuttal, first revision:

Reviewer #1 (Remarks to the Author):

I again want to congratulate the authors to their achievement, which represents a major advancement to the field. With its revision the quality of the manuscript has markedly improved and most issue that I previously raised were appropriately addressed. However, 2 major and a couple of minor issues remain that I would still urge to revise before publication.

We thank the reviewer for his/her constructive review and kind words. We have addressed all remaining concerns, as outlined below.

Major issues not (satisfactorily) addressed:

1)

Original issue: The "mtRNase P field" currently suffers from the use of two different nomenclatures for the 3 subunits ...

Authors reply: "We agree with the reviewer that the names TRMT10C, SDR5C1 and PRORP are concise and consistent with standardized naming. However, we feel that this nomenclature may be confusing to the broad non-expert readership of this paper, and would thus favor MRPP1-3 throughout the text. However, we have clearly introduced the nomenclature in the introduction and also include both nomenclatures in Figure 1. We think this is a good compromise between ensuring consistent naming in the literature and readability of the manuscript, and hope the reviewer agrees."

I honestly don't see why the use of TRMT10C, SDR5C1 and PRORP should be more confusing to the broad non-expert readership than the use of MRPP1-3. On the contrary, it is the continued use of two different nomenclatures for the 3 proteins that will contribute to confusing a broad non-expert readership. The fact that TRMT10C, SDR5C1 and PRORP are concise, consistent with standardized naming, reflect the evolutionary relationship and primary functional significance of each subunit, and that they are approved by the HGNC, none of which applies to MRPP1-3, should actually be sufficient reason to use these names. Occasionally mentioned PRORP orthologues are already now clearly distinguished in the text by specifying them as "single-subunit PRORPs" or by specification of the organism. In the case of the frequently mentioned PRORP1 from *Arabidopsis thaliana*, At could be added (AtPRORP1) whenever the full indication of the species name is not possible or appears inappropriate.

*We have modified both text and figures to adopt the canonical naming (TRMT10C, SDR5C1 and PRORP), as suggested. For *Arabidopsis thaliana* PRORP1, we have introduced the abbreviation At-PRORP1.*

2)

Original issue: Nucleotide (base) positions in tRNAs are by general convention not numbered by simple counting from the 5' end, but by their canonical structural position ...

Authors reply: "We thank the reviewer for pointing this out and agree that numbering should follow the general convention. Unfortunately, structural models in PDB/mmCIF format require consecutive numbering of residues for proper linkage and display, and we therefore cannot adopt this numbering in the models. We have therefore revised the nucleotide position labels in figures and text throughout the manuscript to contain the numbering according to our construct followed by the according canonical tRNA numbering in parenthesis. Furthermore, all references to general tRNA positions have been revised to follow canonical numbering. This way, the reader can both deduce the canonical numbering while reading the manuscript and also easily cross-reference with the numbering in the deposited structural model."

I think the proposed "dual" numbering (consecutive numbering followed by canonical numbering in parenthesis) now used in manuscript and figures will be highly confusing to most readers and not helpful at all, besides the fact that the use of non-canonical numbers when referring to certain nucleotide positions within a tRNA is extremely odd. I would urge to strictly stick to the canonical numbering ONLY throughout the manuscript and all figures, and possibly mention the issue of the non-canonical, consecutive numbering in the deposited models, when referring to them (section "Data availability", line 613); a table translating the consecutive numbering of mitochondrial tRNA-Tyr to its canonical numbering could be provide with the supplements; the information could also be added as a database comment or else with the deposited models. Those models will anyway be consulted by rather few experts only. The inability to canonically number the nucleotides of a tRNA in the models in PDB/mmCIF format does not preclude the use of the canonical numbering in the paper and its figures, as, e.g., demonstrated by the report of the crystal structure of the bacterial ribonuclease P holoenzyme in complex with tRNA (10.1038/nature09516).

We have adopted the canonical numbering for the pre-tRNA^{Tyr} construct throughout. As suggested by the reviewer, we provide a table that cross-references the canonical numbering to the numbering used in the structure as Supplemental Notes 1. We have also described this in the Methods section.

Minor issues:

line 32: Ref. 2 is not appropriate here. If an original reference in addition to the review (ref. 1) is to be provided, then the 3 seminal papers from 1981 would be appropriate here (10.1038/290457a0; 10.1038/290470a0; 10.1038/290465a0).

We have replaced reference 1 with the suggested references.

line 35: Reference to the original papers that first demonstrated the concept (tRNA punctuation) would be appropriate here (same as above: 10.1038/290457a0; 10.1038/290470a0; 10.1038/290465a0).

We have added these references here.

lines 38-42: I am afraid the authors misunderstood my suggestions with respect to the references. Ref. 7 doesn't appear to fit to the first sentence (line 39); ref. 8, 9 could be used here without the need to repeat them then with the next sentence. Ref. 7 would make sense together with refs. 10, 11 in line 42.

We have reorganized these citations as suggested.

line 46: Add ref. 17 here, as in ref. 4 alone only the MRPP1-3 names can be found and the in the following mentioned methylation activity was also only later demonstrated, with ref. 17.

We have added the respective citation as suggested.

line 52 (ref. 14): Maybe better refer to another review instead (e.g., 10.1021/acs.biochem.8b01047) that specifically addresses the enzyme family rather than the modification m1A.

We have replaced the citation with the suggested one.

lines 52-3: “MRPP1 consists of an N-terminal domain (NTD) required for pre-tRNA processing and a dual-specificity C-terminal methyltransferase-domain responsible for establishing the conserved m1G/A methylation at position 9 of mitochondrial tRNAs.” is misleading/wrong. As demonstrated by the authors structure, both domains are apparently required for pre-tRNA processing AND methylation; also previous data did not suggest such split role.

We have rephrased this sentence to:

„TRMT10C consists of an N-terminal domain (NTD) and a dual-specificity C-terminal methyltransferase-domain, which are both required for establishing the conserved m1G/A methylation at position 9 of mitochondrial tRNAs“.

line 56-7: “In contrast to other TRMT10 enzymes which act as monomers or homodimers, ...” As far as I am aware of and in contrast to other SPOUT methyltransferases, all characterized TRM10 enzymes are monomers (see 10.1021/acs.biochem.8b01047; claims by ref. 15 that TRMT10C possibly forms a dimer are finally proven wrong by this paper).

We have rephrased this sentence to:

“In contrast to other TRMT10 enzymes which act as monomers, TRMT10C requires the second subunit of mtRNase P, SDR5C1, for efficient pre-tRNA methylation.”

line 77: Add ref. 3, where the hierarchical order of processing was first shown.

We have added this reference, as suggested.

line 126: “... was not visible in the ...” Maybe better “... was not included in the ...”, because the crystalized protein started only from residue 203.

We have rephrased this sentence as suggested.

line 187 (ref. 15): better cite the original paper (ref. 33).

*Reference 15 (reference 19 in the revised manuscript) is an original paper showing that TRMT10C Q226 is essential for methyltransferase activity (see also our response to the point below). Nevertheless, we have added the suggested reference to the existing one, because it shows that this residue is invariant and also essential for activity of the homologue in *S.pombe*.*

lines 346-48: As far as I am aware of, Q226 has not been studied in TRMT10C, but its mention in ref. 15 (a review) refers to the analysis of the homologous residue in *S. pombe* TRM10 (ref. 33). The sentence and its refs. thus require slight revision.

We believe the reviewer may have confused reference 15 (which is an original research paper by Oerum et al.) with reference 14 (which is a review by Oerum et al., the reference to which we have replaced with a reference to 10.1021/acs.biochem.8b01047 in the revised manuscript, as suggested by the reviewer above). In reference 15 (reference 19 in the revised manuscript), Oerum et al. show that mutation of Q226 to Alanine abolishes methyltransferase activity together with SDR5C1 for both m1A and m1G (Figure 4D). We have therefore kept this sentence as in the reviewed version of the manuscript.

Reviewer #3 (Remarks to the Author):

The authors have substantially improved the manuscript. It is my view that the reviewers' comments and concerns are sufficiently addressed and I enthusiastically recommend this work for publication in NSMB, and believe it would be very well received by the community.

We thank the reviewer for his/her kind words, and have implemented their suggestions as outlined below.

There are a few cosmetic issues that emerged upon re-reading of the revised ms.

1. Line 120. "nicotine-amide dinucleotide (NADH)" seems to be inaccurate. The "A" in NADH refers to adenine, not amide. The full name should read "Nicotinamide adenine dinucleotide".

We thank the reviewer for pointing out this mistake, which we have corrected accordingly.

2. Ex_data Fig. 2b. The arrowhead strikes through the text "RELION".

We thank the reviewer for pointing this out. We have revised this figure and fixed the mistake.

3. Text labels in several figures can be hard to see clearly, esp. In Fig. 2b. I think this is caused by the texts having a fairly thick white outline. I think reducing the outline width should improve text visibility and readability.

We agree with the reviewer and have reduced the outline to improve readability.

4. Fig. 3a, “m-tRNATyr” label should read “mt-tRNATyr”. The “t” is missing.

We thank the reviewer for pointing this out and have corrected the figure accordingly.

5. Fig. 4c. The two-colored text label for “MRPP1 methyltransferase domain” seems unnecessary, perhaps to indicate the “chimeric” nature of the structural model? I find it distracting and think a single color should work well, or simply omitting the same label already shown in panel b.

We agree with the reviewer and have chosen to show the motif II loop in grey only to emphasize that it is derived from the previous crystal structure. We have additionally clearly stated this in the corresponding figure legend.

6. Fig. 4b, c. I think it would help guide non-expert readers to explicitly label N1, the site of methylation.

We agree with the reviewer and have labeled N1 accordingly.

Final Decision Letter:

1st Jul 2021

Dear Hauke,

We are now happy to accept your revised paper “Structural basis of RNA processing by human mitochondrial RNase P” for publication as an Article in Nature Structural & Molecular Biology.

Before the manuscript is sent to the printers, we shall make any detailed changes in the text that may be necessary either to make it conform with house style or to make it intelligible to a wider readership. If the changes are extensive, we will ask for your approval before the manuscript is laid out for production. Once your manuscript is typeset you will receive a link to your electronic proof via email within 20 working days, with a request to make any corrections within 48 hours. Please read proofs with great care to make sure that the sense has not been altered. If you have queries at any point during the production process then please contact the production team

at rjsproduction@springernature.com. Once your paper has been scheduled for online publication, the Nature press office will be in touch to confirm the details.

Please note that due to tight production schedules, proofs should be returned as quickly as possible to avoid delaying publication. If you anticipate any limitations to your availability over the next 2-4 weeks (such as vacation or traveling to conferences, etc.), please e-mail rjsproduction@springernature.com as soon as possible. Please provide specific dates that you will be unavailable and provide detailed contact information for an alternate corresponding author if necessary.

As soon as your article is published, you can generate your shareable link by entering the DOI of your article here: http://authors.springernature.com/share. Corresponding authors will also receive an automated email with the shareable link

Your paper will be published online soon after we receive proof corrections and will appear in print in the next available issue. You can find out your date of online publication by contacting the production team shortly after sending your proof corrections. Content is published online weekly on Mondays and Thursdays, and the embargo is set at 16:00 London time (GMT)/11:00 am US Eastern time (EST) on the day of publication. Now is the time to inform your Public Relations or Press Office about your paper, as they might be interested in promoting its publication. This will allow them time to prepare an accurate and satisfactory press release. Include your manuscript tracking number (NSMB-A44742B) and our journal name, which they will need when they contact our press office.

About one week before your paper is published online, we shall be distributing a press release to news organizations worldwide, which may very well include details of your work. We are happy for your institution or funding agency to prepare its own press release, but it must mention the embargo date and Nature Structural & Molecular Biology. If you or your Press Office have any enquiries in the meantime, please contact press@nature.com.

If you have not already done so, we strongly recommend that you upload the step-by-step protocols used in this manuscript to the Protocol Exchange. Protocol Exchange is an open online resource that allows researchers to share their detailed experimental know-how. All uploaded protocols are made freely available, assigned DOIs for ease of citation and fully searchable through nature.com. Protocols can be linked to any publications in which they are used and will be linked to from your article. You can also establish a dedicated page to collect all your lab Protocols. By uploading your Protocols to

Protocol Exchange, you are enabling researchers to more readily reproduce or adapt the methodology you use, as well as increasing the visibility of your protocols and papers. Upload your Protocols at www.nature.com/protocolexchange/. Further information can be found at www.nature.com/protocolexchange/about.

Please note that *Nature Structural & Molecular Biology* is a Transformative Journal (TJ). Authors may publish their research with us through the traditional subscription access route or make their paper immediately open access through payment of an article-processing charge (APC). Authors will not be required to make a final decision about access to their article until it has been accepted. [Find out more about Transformative Journals](https://www.springernature.com/gp/open-research/transformative-journals)

Authors may need to take specific actions to achieve compliance with funder and institutional open access mandates. For submissions from January 2021, if your research is supported by a funder that requires immediate open access (e.g. according to [Plan S principles](https://www.springernature.com/gp/open-research/plan-s-compliance)) then you should select the gold OA route, and we will direct you to the compliant route where possible. For authors selecting the subscription publication route our standard licensing terms will need to be accepted, including our [self-archiving policies](https://www.springernature.com/gp/open-research/policies/journal-policies). Those standard licensing terms will supersede any other terms that the author or any third party may assert apply to any version of the manuscript.

With kind regards,
Anke

Anke Sparmann, PhD
Senior Editor
Nature Structural and Molecular Biology
ORCID 0000-0001-7695-2049
